# A systematic literature review on the applications of recurrent neural networks in code clone research

**Fahmi H. Quradaa**[1,2]* , **Sara Shahzad**[1], **Rashad S. Almoqbily**[1,2]

**1** Department of Computer Science, University of Peshawar, Peshawar, Pakistan, **2** Department of Computer Science, Aden Community College, Aden, Yemen

☯ These authors contributed equally to this work.

* qurada@uop.edu.pk

**Data Availability Statement:** All relevant data are within the paper and its Supporting Information file.

**Funding:** The author(s) received no specific funding for this work.

## Abstract

Code clones, referring to code fragments that are either similar or identical and are copied and pasted within software systems, have negative effects on both software quality and maintenance. The objective of this work is to systematically review and analyze recurrent neural network techniques used to detect code clones to shed light on the current techniques and offer valuable knowledge to the research community. Upon applying the review protocol, we have successfully identified 20 primary studies within this field from a total of 2099 studies. A deep investigation of these studies reveals that nine recurrent neural network techniques have been utilized for code clone detection, with a notable preference for LSTM techniques. These techniques have demonstrated their efficacy in detecting both syntactic and semantic clones, often utilizing abstract syntax trees for source code representation. Moreover, we observed that most studies applied evaluation metrics like F-score, precision, and recall. Additionally, these studies frequently utilized datasets extracted from open-source systems coded in Java and C programming languages. Notably, the Graph-LSTM technique exhibited superior performance. PyTorch and TensorFlow emerged as popular tools for implementing RNN models. To advance code clone detection research, further exploration of techniques like parallel LSTM, sentence-level LSTM, and Tree-Structured GRU is imperative. In addition, more research is needed to investigate the capabilities of the recurrent neural network techniques for identifying semantic clones across different programming languages and binary codes. The development of standardized benchmarks for languages like Python, Scratch, and C#, along with cross-language comparisons, is essential. Therefore, the utilization of recurrent neural network techniques for clone identification is a promising area that demands further research.

## 1. Introduction

Software bad smells point to potential issues within systems and have a detrimental impact on software quality [1]. Among these issues, code clones, similar or identical fragments resulting

**Competing interests:** The authors have declared that no competing interests exist.

from copying and pasting existing code, stands out as a significant concern [2]. While code cloning may accelerate the development process, it ultimately hampers software quality and maintainability [3, 4]. This phenomenon leads to bug propagation, update irregularities, inflated codebases, and compromised software architecture [5]. Consequently, the detection of clones has emerged as a vibrant research field with encompassing a variety of techniques and tools [6–9].

Recent breakthroughs in machine learning (ML), particularly within the domains of language modeling [10], machine translation [11], speech recognition [12], and online handwritten [13], have sparked interest of researcher in leveraging ML algorithms for the detection of both syntactic and semantic clones. A diverse array of techniques have been applied to learn distinct patterns that distinguish clones from non-clones, even across different clone categories [14, 15]. Deep learning (DL) techniques, like recurrent neural networks (RNNs), graph neural networks (GNN), and others have proven effective in capturing both syntactic and semantic clones [14]. The advancement of RNNs, as highlighted by Alex Graves [16], demonstrates their proficiency as sequential learners capable of capturing features and long-term dependencies.

Several systematic literature reviews (SLRs) have been carried out in the field of code clones. Roy and Cordy [7] conducted a comprehensive survey of software clone detection techniques up until 2006, covering definitions, types, benefits, drawbacks, and applications of code clone detection. They categorized techniques into the following categories: text-based, token-based, tree-based, graph-based, metrics-based, and hybrid. Notably, it excluded modern ML approaches. Rattan et al. [6] explored various categories of clone types and emphasized the significance of identifying semantic and model clones. However, they did not cover ML clone detection techniques. Al-Shaaby et al. [17], Azeem et al. [3], and Al-Azba et al. [18] surveyed ML and DL algorithms for code smells detection, but their primary focus was not on clone detection or RNN techniques. Qurat et al. [8] reviewed clone detection techniques/tools spanning from 2013 to 2018, mentioning ML techniques used in clone detection but without in-depth coverage of specific techniques. Likewise, Shobha et al. [9] discussed detection techniques and emphasized the need for innovative methods capable of simultaneously identifying all four types of clones. However, they did not explore the utilization of ML techniques for detecting code clones. Maggie et al. [14] performed a systematic review of DL techniques for detecting code clones. They reviewed recent DL techniques, analyzing their effectiveness, complexity, and scalability, and addressing associated challenges and limitations. Kaur and Rattan [15] presented a systematic review of ML techniques in code clone detection up to 2020, providing insights into clone types, metrics, datasets, code representation, and tools. Morteza et al. [19] conducted a systematic review on code similarity measurement and clone detection, offering insights into the current techniques, applications, and challenges. However, their review did not specifically emphasize advancement in RNN in the field of code clone research.

This study aims to comprehensively review recent scientific research between 2015 and 2022 on detecting code clones using RNN techniques. We followed the established guidelines outlined by Kitchenham and Charters [20], we extensively searched seven digital databases with a predefined search string. Then, we applied a set of inclusion/exclusion criteria, along with quality assessment criteria. Moreover, we utilized a snowballing process to identify any relevant studies that might have been missed. The data extracted from the selected studies were analyzed and compared based on several factors, including the types of RNN applications, the accuracy of techniques, the type of detected clones, techniques used for code representation, and the datasets used.

We believe that the review's findings are highly valuable for software developers, practitioners, and researchers in software clone analysis. They will acquire essential insights into

prevalent RNN applications for clone detection, aiding developers in selecting optimal RNN applications for their goals. The findings also shed light on the accuracy metrics used to evaluate the models. Additionally, researchers can use these results as a valuable reference and identify new research challenges in clone detection using RNN applications. Moreover, The study offers a detailed analysis of RNN applications in code clone detection.

The contributions of our systematic literature review (SLR) can be summarized as follows: Firstly, it identifies and categorizes studies between 2015 and 2022 that use RNN techniques in code clone detection. Secondly, it analyzes the selected studies to provide valuable information to the research community regarding (1) the applied RNN techniques in clone detection research, (2) the types of clones detected, (3) the evaluation metrics used to assess the performance of these techniques, (4) the code representation techniques used, (5) the datasets used in the experiments to train and test RNN models, and (6) the most frequently used tools to building RNN models. Lastly, it offers recommendations and identifies significant gaps for future research in this field.

The study is structured as follows: Section 2 provides a background on code clones and recurrent neural networks (RNNs). Section 3 outlined the research methodology. Results are presented in Section 4, and the research question are discussed in Section 5. Potential threats to the validity of the study are noted in Section 6, and Section 7 concludes and suggests future work.

## 2. Background

In this section, we present a brief background on code cloning and recurrent neural network (RNN) techniques.

### 2.1 Code clones

Code clone detection techniques generally involve three key steps: (1) Code pre-processing, which eliminates uninteresting items like header files, and comments; (2) Code representation, where source code is transformed into an intermediate representation (e.g., AST, token sequence, or PDG); and (3) Code similarity comparison: where code fragments similarity is computed, leading to identification of code clones when this similarity exceeds a predefined threshold.

Researchers often categorize code clones into four distinct types [6, 21]. The first three types (Type-I, Type-II, and Type-III) primarily revolve around textual similarities, while the fourth type (Type-IV) places a greater emphasis on functional or semantic resemblances. Type-I clones encompass code fragments that are essentially identical, differing only in minor aspects such as comments, formatting, and whitespaces. Type-II clones involve code fragments that are syntactically identical, except for minor disparities in identifier names, variables, formatting, data types, and literals. In the case of Type-III clones, beyond the variations encountered in Type-II, the code fragments are syntactically equivalent with additional adjustments, such as inclusion or omissions of statements. Contrastingly, Type-IV clones exhibit substantial dissimilarities both in text and syntax while performing equivalent functionalities. Between Type-III and Type-IV, there exists a range of clones that, despite retaining certain syntactic resemblances, present significant challenges in their detection. These are often referred to as clones in the Twilight zone [22]. Clones in Twilight zone are further classified into the following subtypes: Very Strongly Type-III (syntactic similarity between 90% and less than 100%), Strongly Type-III (syntactic similarity between 70% and less than 90%), Moderately Type-III (syntactic similarity between 50% and less than 70%), and Weakly Type-III/Type-IV (syntactic similarity of clone less than 50%).

## 2.2 Recurrent Neural Networks (RNNs)

RNNs techniques are used in a wide array of fields, including research in software code cloning [23]. In this study, we classify RNN techniques for clone detection into three distinct categories: traditional recurrent neural networks (TRNN), Long Short Term Memory (LSTM), and gated recurrent units (GRU). Traditional RNNs are composed of two main types: original recurrent neural networks (ORNNs) and bidirectional recurrent neural network (Bi-RNNs). ORNN adapts the structure of feed-forward neural networks to handle variable-length sequences, utilizing self-connections to retain memory [23]. However, they encounter challenges like vanishing and exploding gradient problems [24]. Bi-RNNs, introduced by Schuster and Paliwal [25], overcomes these ORNNs limitations by incorporating two independently hidden layers that process input sequences in both forward and backward directions. The LSTM category, known for its effectiveness in mitigating vanishing gradients problems, incorporates memory cells and gates to regulate data flow. Specific techniques within the LSTM category include original LSTM (OLSTM) [26], Bidirectional LSTM (Bi-LSTM) [27], and Tree-LSTM [28]. OLSTM employs input, output, and forget gates [29]. Bi-LSTM combines forward and backward LSTM layers for enhanced performance [24, 27, 30]. Tree-LSTM processes structured input through tree-based units, with Child-Sum Tree-LSTM designed for high branching factor trees, and Binary Tree-LSTM suited for binary trees. Graph-LSTM [31] adapts the Tree-LSTM architecture for processing graph data, incorporating syntax and semantics while preserving the original structures. Nevertheless, LSTM architectures are significantly more complex in the hidden layer, resulting in approximately four times more parameters than a simple RNN architecture [32]. In contrast, the GRU category offers a simpler alternative to LSTM. It includes only two gating units to address vanishing/exploding gradient problems in handling long-term dependencies. Unlike LSTM's three gating units, GRU uses reset and update gates. The GRU category offers two variations [33, 34]: original GRU (OGRU), which manages new inputs and state retention, while Bidirectional GRU (Bi-GRU) employing two layers for forward-backward processing. This results in faster training and improved information integration.

## 3. Review methodology

This systematic review aims to analyze the existing studies on the usage of RNNs in code clone detection, in accordance with established guidelines and protocols defined by Kitchenham and Charters [20, 35–37]. A systematic literature review (SLR) is a rigorous method for assessing and interpreting relevant research [38–41], providing a strong foundation for making claims. Unlike ad hoc reviews, SLRs require more effort [42]. The SLR process involves creating a review protocol, conducting the review, reporting findings, and discussing results. Our review protocol includes research questions, search strategies, criteria for inclusion/exclusion, quality assessment, and data synthesis. Please refer to Fig 1 for an overview of PRISMA flow chart for the selection process and Fig 2 for an overview of SLR steps.

### 3.1 Research questions

Defining essential research questions is vital for the review process. We have formulated specific questions aligned with our objective. Listed in Table 1 with their motivations.

### 3.2 Search strategy and study resources

We developed a search strategy to collect essential studies that address the review's research questions. This process involves identifying search terms, defining a global search string, and selecting digital study resources to search through.

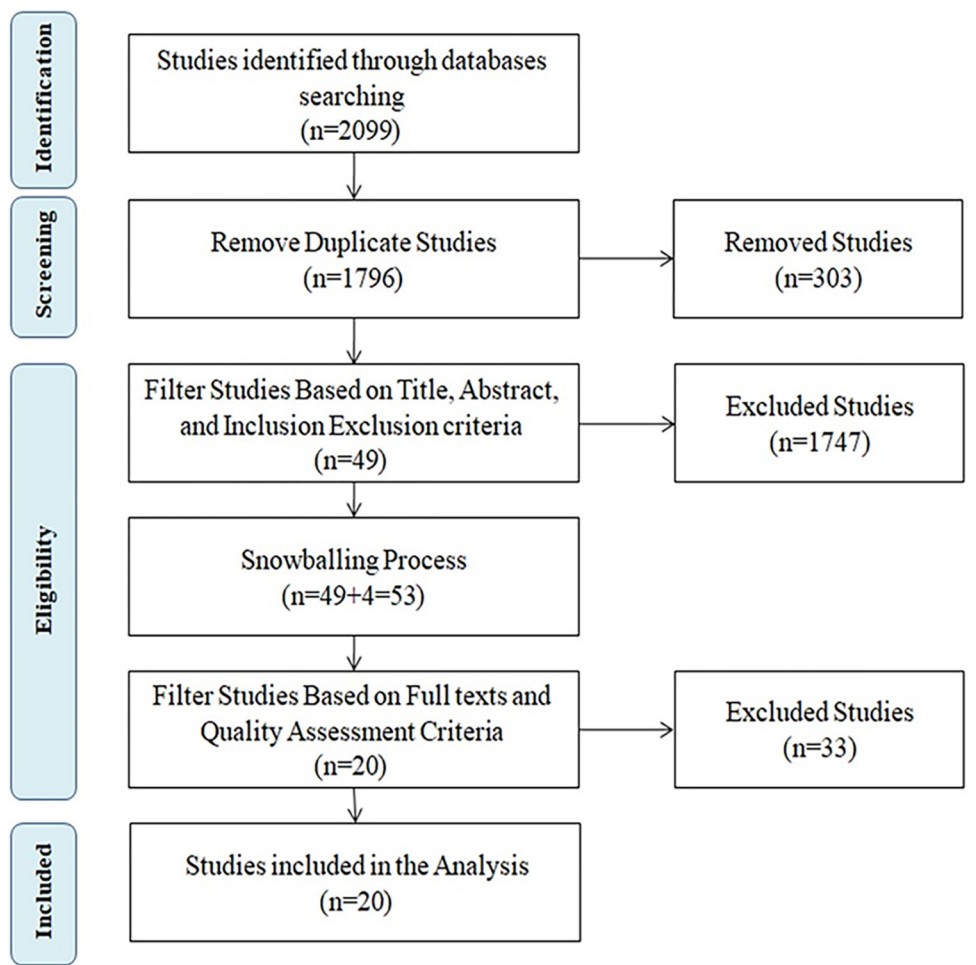

**Fig 1. PRISMA flow diagram outlining the study selection process.**

**3.2.1 Identifying search terms and defining global search string.** In this step, we generated search terms for our research questions and created a search string. This string is used to query online databases and collect relevant articles that address the research questions of the review. The process comprises the following steps: Firstly, we identified the major terms from the research questions. Then, we created a list of synonyms and alternative spellings for each major term. To ensure accuracy, we cross-referenced this list with keywords in relevant papers. Next, we linked each major term with its synonyms and alternative spellings using the Boolean OR operator. Lastly, we refined the search by combining the major terms with the Boolean AND operator. Applying these steps resulted in the following search string:

*((("Recurrent neural network" OR "Bi- Recurrent neural network" OR "Long short term memory" OR \*LSTM OR "Gated Recurrent Units" OR GRU) AND (code OR application OR software) AND (copy OR clone OR cloning OR duplicat\* OR similar\*)).*

Because of constraints in search terms and Boolean operators within specific digital libraries (Science direct), we defined a shorter search string as follows:

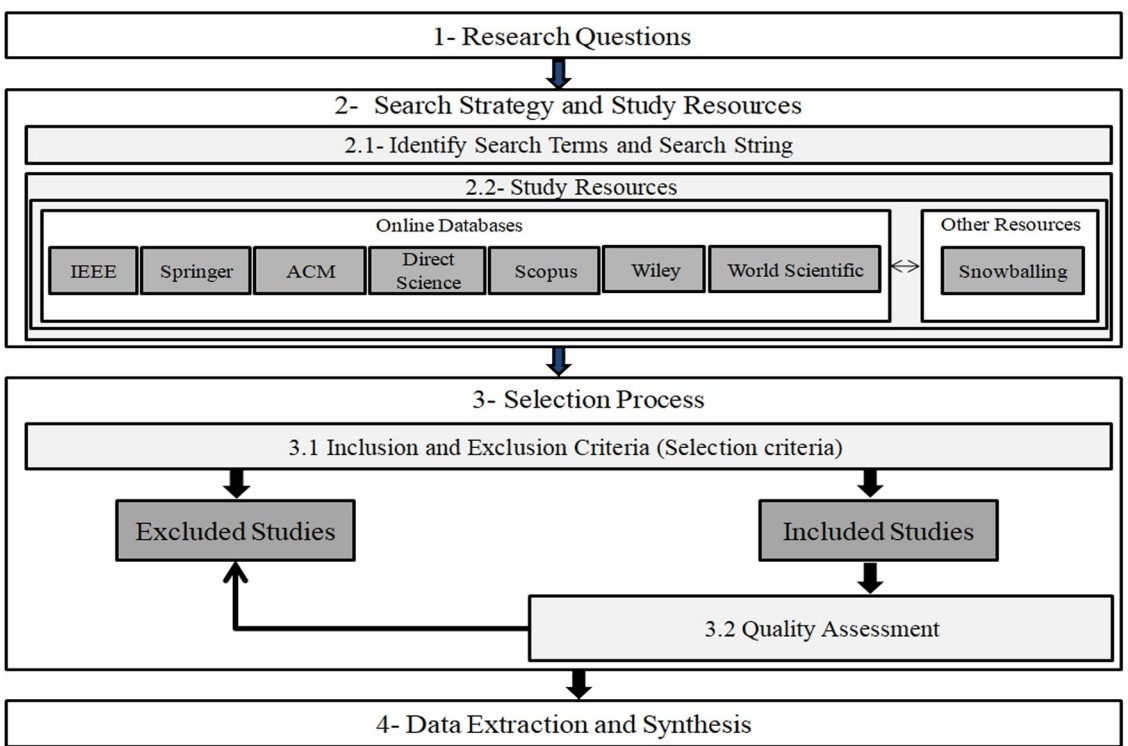

**Fig 2. Overview of systematic literature review.**

**Table 1. Research questions and main motivations.**

| Research Question | Main motivation |
|---|---|
| **RQ1**: What RNN techniques have been employed for code clone detection? | The results assist practitioners and researchers in identifying popular RNN techniques for code clone detection and encourage exploration of unused techniques. |
| **RQ2**: What types of clones are most frequently detected using RNN techniques? | To identify code clones detected by RNN techniques, researchers can explore less-studies clone types. |
| **RQ3**: What source code representation techniques have been used in RNN applications? | Identifying source code representation techniques in RNN can reveal commonly used and overlooked techniques. |
| **RQ4**: What datasets were used in the selected studies? | By examining dataset characteristics including their name, type (commercial, student, open source), availability (online availability), and language, researchers can gain valuable insights for reusing or creating datasets. |
| **RQ5**: What are the most commonly used tools for building RNN models for clone detection? | To explore the tools used for implementing RNN models, researchers can better select the most suitable tool to meet their needs. |
| **RQ6**: What evaluation metrics are used to assess the effectiveness of RNN models? | Identifying performance metrics for RNN models aids researchers in choosing appropriate accuracy measures. |
| **RQ7**: Which RNN technique yields superior outcomes when evaluated on the same dataset for the same problem? | To compare and select the best RNN technique, researchers can assess performance across various studies. |

**Table 2. Selected online databases.**

| # | Database Name | URL |
|---|---|---|
| 1 | ACM Digital Library | https://dl.acm.org/ |
| 2 | IEEE Xplore Digital Library | https://ieeexplore.ieee.org |
| 3 | Science Direct Library | https://www.sciencedirect.com/ |
| 4 | Scopus Digital Library | https://www.Scopus.com/ |
| 5 | Springer Link Digital Library | https://link.springer.com/ |
| 6 | Wiley Digital Library | https://onlinelibrary.wiley.com/ |
| 7 | World Scientific | https://www.worldscientific.com/ |

*("Recurrent neural network" OR LSTM OR GRU) AND (code OR application OR software) AND (clone OR duplicate OR similar))*

**3.2.2 Study resources.** In order to increase the chances of finding highly relevant journal and conference articles using our search string, we opted to use seven reputable online scientific databases. These databases are renowned for publishing research on deep learning applications and code clone detection techniques. Additionally, Kitchenham et al. and Brereton et al [35–37] endorse electronic database searches. Refer to Table 2 for the selected databases.

Furthermore, we conducted a snowballing process [43] to identify additional sources by reviewing the references and citations of relevant studies. This approach expanded our selection and minimized the risk of overlooking any relevant articles.

## 3.3 Selection process

The SLR search initiates with separate searches in the digital libraries specified in Table 2, employing the predefined search string as detailed in Section 3.2.1. Our database search was performed using the advanced search functionality, which allowed us to refine our search using different options, as shown in Table 3. Furthermore, we restricted publication date of the included studies to fall between 2015 and 2022. The search process continued without date restrictions until March 2023.

The Results of this search are presented in Table 3 and Fig 3, revealing a total of 2099 candidate/initial studies. However, a significant portion of these studies lacks pertinent information needed to address the research questions. Consequently, the process of filtering out unqualified studies becomes essential to identify the studies that are relevant to our objectives.

Before initiating the filtering process, we removed duplicate studies from the initial list of potential candidate studies. During the sorting step, a total of 303 duplicate studies were detected within the digital libraries and subsequently removed.

**Table 3. Summary of search process from different digital libraries.**

| # | Digital Database | Search String in | Content type | Subjects | #studies |
|---|---|---|---|---|---|
| 1 | ACM Digital Library | Abstract | All sources | All subjects | 443 |
| 2 | IEEE Xplore Digital Library | Abstract | All sources | All subjects | 397 |
| 3 | Science Direct Digital Library | Title,abstract,keywords | All sources | All subjects | 93 |
| 4 | Scopus Digital Library | Abstract | All sources | All subjects | 434 |
| 5 | Springer Link Digital Library | All | Jour&conf | Computer science | 286 |
| 6 | Wiley Digital Library | Abstract | All sources | All subjects | 377 |
| 7 | World Scientific | Abstract | All sources | All subjects | 69 |
| | | | | | **Total: 2,099** |

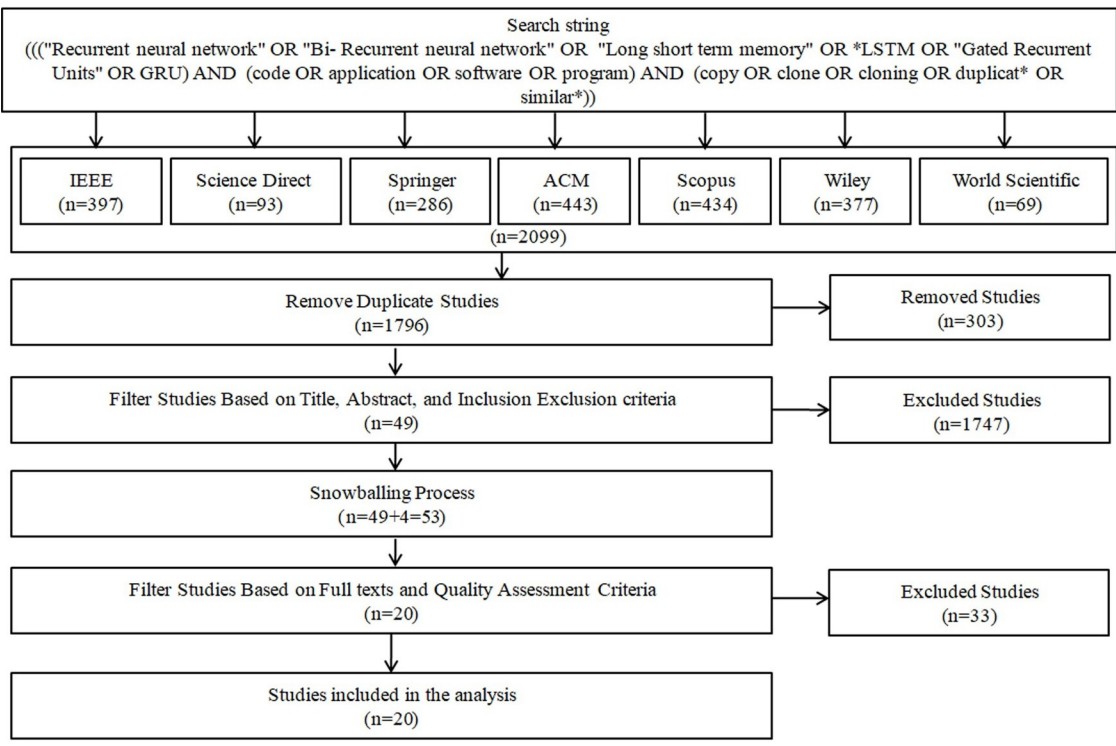

**Fig 3. Study selection process.**

Two filtering rounds are performed to assess the relevance of candidate/initial studies on the utilization of RNN techniques for code clones detection [44]. In the first round, two authors independently evaluate the titles, abstracts, and keywords of each study, applying the inclusion and exclusion criteria outlined in Section 3.3.1. Any disagreements among the authors were addressed through discussions in meetings and subsequently resolved.

**3.3.1 Inclusion and exclusion criteria.** To be included in the SLR, studies must satisfy the following inclusion criteria.

- All studies that have used any RNN technique and belong to the code clone detection research area.

- Studies published in journals, conference proceedings, and workshops.

- Studies should be published in 2015 to 2022.

- Studies should be at least 6 pages long.

Conversely, the following exclusion criteria are applied to exclude unqualified studies from the SLR:

- Any study within the code clone research area that does not use RNN techniques.

- Non-per-reviewed studies (e.g., those available on platforms like arXiv.org).

- Studies that are not written in English language.

- If a study is published in conference and journal then, the expanded version is chosen.

**Table 4. Quality assessment criteria.**

| Criterion | Quality Questions | Yes | No | Partly |
|-----------|-------------------|-----|-----|--------|
| Q1 | Is the study using any RNN techniques? | | | |
| Q2 | Is the study discussing code clone detection? | | | |
| Q3 | Are the objectives of the study clearly stated? | | | |
| Q4 | Is the experiment's design well-defined? | | | |
| Q5 | Is there a clear statement of findings? | | | |
| Q6 | Is there a detailed description of the study's performance metrics? | | | |
| Q7 | Is the tool used to implement the RNN model reported? | | | |
| Q8 | Does the study compare the proposed work with other techniques? | | | |
| Q9 | Does the study clearly define the source code representation method? | | | |
| Q10 | Does the study address validity concerns or limitations? | | | |
| Q11 | Does the study mention the type of clone? | | | |
| Q12 | Is the dataset clearly and sufficiently described? | | | |

Following the first round of filtering, a total of 1747 studies were excluded, resulting in the selection of only 49 studies, as shown in Fig 3. Subsequently, a snowballing process [43] was conducted by examining the reference lists of these 49 relevant studies. These process led to the discovery of 4 additional studies: 3 from IEEE and 1 from ACM, resulting in a total of 53 relevant studies. During the second filtering round, we examined the 53 relevant studies identified in the first round to identify those studies addressing our SLR research questions. We comprehensively reviewed the full texts and applied the quality assessment criteria outlined in Section 3.3.2.

**3.3.2 Quality assessment criteria.** In the second round, we establish a set of quality assessment criteria that are aligned with our SLR objectives, which we used to evaluate the quality of the selected studies. The specific criteria are detailed in Table 4, with some of them being adapted from Dyba et al. [45] and Singh et al. [46].

We assess the quality of the selected studies through the approach detailed in references [17, 45, 47]. Quality is evaluated with Yes, No, or Partially (inferred from text) responses, scored as follows: Yes = 1, No = 0, and Partly = 0.5. The study's overall quality is determined by the sum of the scores from 12 questions. Studies scoring $\geq 6$ are selected, as outlined in reference [47]. Table 5 shows the final selected studies and their corresponding scores. During this step, two authors conducted independent assessment of the selected studies. In case where disagreements arose, they were addressed and resolved through discussions involving the third author.

Both filtering rounds resulted in the selection of 20 studies. The details of these 20 studies can be found in Table 6. Scores for quality assessment of the relevant studies are listed in S1 Appendix.

For easy tracking, we assigned an ID to each study and marked it accordingly. The detailed selection process is shown in Fig 3. Furthermore, we used Endnote software to manage and gather studies throughout the search process.

## 3.4 Data extraction and synthesis

After identifying the appropriate studies, we proceeded to extract and document essential data that relevant to our research questions. To facilitate this process, we developed a dedicated data extraction template for capturing the study details. The data extraction template employed for recording this information is outlined in Table 7.

**Table 5. Final selected studies and quality assessment scores.**

| ID | Ref | Q1 | Q2 | Q3 | Q4 | Q5 | Q6 | Q7 | Q8 | Q9 | Q10 | Q11 | Q12 | Score |
|----|-----|----|----|----|----|----|----|----|----|----|-----|-----|-----|-------|
| S1 | [48] | 1 | 1 | 1 | 1 | 1 | 1 | 0 | 1 | 1 | 0 | 1 | 1 | 10 |
| S2 | [49] | 1 | 1 | 1 | 1 | 1 | 1 | 1 | 1 | 1 | 1 | 1 | 1 | 12 |
| S3 | [50] | 1 | 1 | 1 | 1 | 1 | 1 | 1 | 1 | 1 | 1 | 0.5 | 1 | 11.5 |
| S4 | [51] | 1 | 1 | 1 | 1 | 1 | 1 | 1 | 0 | 1 | 0 | 0.5 | 1 | 9.5 |
| S5 | [52] | 1 | 1 | 1 | 1 | 1 | 1 | 1 | 1 | 1 | 1 | 0.5 | 1 | 11.5 |
| S6 | [53] | 1 | 1 | 1 | 1 | 1 | 1 | 1 | 1 | 1 | 1 | 0.5 | 1 | 11.5 |
| S7 | [4] | 1 | 1 | 1 | 1 | 1 | 1 | 1 | 1 | 1 | 1 | 0.5 | 1 | 11.5 |
| S8 | [54] | 1 | 1 | 1 | 1 | 1 | 1 | 1 | 0 | 1 | 1 | 1 | 1 | 11 |
| S9 | [55] | 1 | 1 | 1 | 1 | 1 | 1 | 1 | 1 | 1 | 1 | 1 | 1 | 12 |
| S10 | [56] | 1 | 1 | 1 | 1 | 1 | 1 | 0 | 1 | 1 | 1 | 0.5 | 1 | 10.5 |
| S11 | [57] | 1 | 1 | 1 | 1 | 1 | 1 | 1 | 1 | 1 | 1 | 0.5 | 1 | 11.5 |
| S12 | [58] | 1 | 1 | 1 | 1 | 1 | 1 | 1 | 1 | 1 | 1 | 1 | 1 | 12 |
| S13 | [59] | 1 | 1 | 1 | 1 | 1 | 1 | 0 | 1 | 1 | 0 | 0.5 | 1 | 9.5 |
| S14 | [60] | 1 | 1 | 1 | 1 | 1 | 1 | 1 | 0 | 1 | 0 | 1 | 1 | 10 |
| S15 | [61] | 1 | 1 | 1 | 1 | 1 | 1 | 1 | 1 | 1 | 1 | 1 | 1 | 12 |
| S16 | [62] | 1 | 1 | 1 | 1 | 1 | 1 | 1 | 1 | 1 | 0 | 0.5 | 1 | 10.5 |
| S17 | [63] | 1 | 1 | 1 | 1 | 1 | 1 | 0 | 1 | 1 | 0 | 1 | 1 | 10 |
| S18 | [64] | 1 | 1 | 1 | 1 | 1 | 1 | 1 | 1 | 1 | 1 | 0.5 | 1 | 11.5 |
| S19 | [31] | 1 | 1 | 1 | 1 | 1 | 1 | 1 | 1 | 1 | 1 | 0.5 | 1 | 11.5 |
| S20 | [65] | 1 | 1 | 1 | 1 | 1 | 1 | 1 | 1 | 1 | 0 | 0.5 | 1 | 10.5 |

**Table 6. The final selected studies.**

| ID | Year | Study Title | Type | Venue |
|----|------|-------------|------|-------|
| S1 | 2020 | A novel code stylometry-BASED code clone detection strategy | Conference | IWCMC |
| S2 | 2019 | A novel neural source code representation based on abstract syntax tree | Conference | ICSE |
| S3 | 2021 | Asteria Deep Learning-based AST-Encoding for Cross-platform Binary Code Similarity Detection | Conference | DSN |
| S4 | 2021 | Bindeep A deep learning approach to binary code similarity detection | Journal | Expert Systems with Applications |
| S5 | 2022 | Crolssim Cross-language software similarity detector using hybrid approach of LSA-based AST-mdrep features and CNN-LSTM model | Journal | Int. Journal of Intelligent Systems |
| S6 | 2019 | Cross-Language Clone Detection by Learning Over Abstract Syntax Trees | Conference | MSR |
| S7 | 2016 | Deep learning code fragments for code clone detection | Conference | ASE |
| S8 | 2018 | Deep Learning Similarities from Different Representations of Source Code | Conference | MSR |
| S9 | 2021 | FCCA Hybrid Code Representation for Functional Clone Detection Using Attention Networks | Journal | IEEE Transactions on Reliability |
| S10 | 2020 | From Local to Global Semantic Clone Detection | Conference | DSA |
| S11 | 2021 | Mulcode: A Multi-task Learning Approach for Source Code Understanding | Conference | SANER |
| S12 | 2020 | Modular Tree Network for Source Code Representation Learning | Journal | ACM Transactions on Software Engineering and Methodology |
| S13 | 2017 | Plagiarism Detection in Programming Assignments Using Deep Features | Conference | ACPR |
| S14 | 2018 | Positive and unlabeled learning for detecting software functional clones with adversarial training | Conference | IJCAI |
| S15 | 2020 | SCDetector software functional clone detection based on semantic tokens analysis | Conference | ASE |
| S16 | 2020 | Siamese-Based BiLSTM Network for Scratch Source Code Similarity Measuring | Conference | IWCMC |
| S17 | 2017 | Supervised deep features for software functional clone detection by exploiting lexical and syntactical information in source code | Conference | IJCAI |
| S18 | 2021 | VDSimilar Vulnerability detection based on code similarity of vulnerabilities and patches | Journal | Computers & Security |

*(Continued)*

**Table 6.** (Continued)

| ID | Year | Study Title | Type | Venue |
|----|------|-------------|------|-------|
| S19 | 2022 | Hierarchical semantic-aware neural code representation | Journal | Journal of Systems & Software |
| S20 | 2021 | Clone detection in 5G-enabled social IoT system using graph semantics and deep learning model | Journal | Int. Journal of Machine Learning and Cybernetics |

The data extraction process involved the following steps: collecting bibliographic details of studies, capturing key findings and techniques, and conducting a comprehensive analysis to address research questions. All authors participated in this process. One author reviewed each study, extracting and recording relevant data using a predefined template. Other authors independently reviewed a random sample of studies and cross-verified the results. In the case of any disagreements among the authors regarding results, consensus meetings were held to resolve the issues. This process is similar to the methods used in prior studies [17, 66–68].

After completing the data extraction, the extracted data is stored in a file for the next step. Table 8 reveals that the final selected studies can address most of the research questions in this

**Table 7. Data extraction template.**

| Item | Description |
|------|-------------|
| Extractor name | The researcher responsible for extracting the data. |
| Extraction date | Date of extraction data. |
| Data checker | [Name of the researcher] verified the extracted data. |
| Identifier | Study identifier. |
| Database | ACM/IEEE/Direct Science/Springer/Wiley/ Scopus/ World Scientific. |
| Title | Title of the Study. |
| Tool/approach name | Name of the proposed tool/approach. List the first authors if not available. |
| Author | The authors of the study. |
| Venue | Publication venue. |
| Year | Publication year. |
| Reference type | journal / conference/ workshop. |
| Study type | Experimental/Case study/ survey/ algorithm. |
| RNN application used | What type of RNN application has been applied in the study? |
| Classification type | What type of classification is being used? binary/ multi-class classification |
| Training strategy | What is the training strategy being used? (supervised, unsupervised, or semi-supervised) |
| Clone types &levels | What types of code clones are detected by the proposed tool/approach, and at what level are they identified? block-level/ method-level/ class-level. |
| Source code representation | What technique is being employed for the source code representation? |
| Evaluation Metrics | Which measures are employed to evaluate the model? Precision/ Recall/ F score/ AUC-ROC. |
| Availability | Is the tool/approach available? If Yes provide the URL. |
| Development tool | Which tool was used to construct the RNN model? |
| Dataset name | Which datasets were utilized for training and testing the model? |
| Dataset size | What is the size of the dataset? |
| Dataset Type | What is the type of the dataset? commercial, student, open source |
| Language | What programming language is used in the dataset? |
| Availability | Is the dataset publically available? Provide URL if Yes. |
| Compared approaches | What approaches were used to compare with the current work? |

**Table 8. Selected studies and addressed research questions.**

|  | S₁ | S₂ | S₃ | S₄ | S₅ | S₆ | S₇ | S₈ | S₉ | S₁₀ | S₁₁ | S₁₂ | S₁₃ | S₁₄ | S₁₅ | S₁₆ | S₁₇ | S₁₈ | S₁₉ | S₂₀ |
|---|---|---|---|---|---|---|---|---|---|---|---|---|---|---|---|---|---|---|---|---|
| RQ1 | √ | √ | √ | √ | √ | √ | √ | √ | √ | √ | √ | √ | √ | √ | √ | √ | √ | √ | √ | √ |
| RQ2 | √ | √ | √ | √ | √ | √ | √ | √ | √ | √ | √ | √ | √ | √ | √ | √ | √ | √ | √ | √ |
| RQ3 | √ | √ | √ | √ | √ | √ | √ | √ | √ | √ | √ | √ | √ | √ | √ | √ | √ | √ | √ | √ |
| RQ4 | √ | √ | √ | √ | √ | √ | √ | √ | √ | √ | √ | √ | √ | √ | √ | √ | √ | √ | √ | √ |
| RQ5 | X | √ | √ | √ | √ | √ | √ | √ | √ | √ | √ | √ | X | X | √ | √ | X | √ | √ | √ |
| RQ6 | √ | √ | √ | √ | √ | √ | √ | √ | √ | √ | √ | √ | √ | √ | √ | √ | √ | √ | √ | √ |
| RQ7 | √ | √ | √ | √ | √ | X | X | X | √ | X | √ | √ | √ | √ | √ | √ | √ | √ | √ | √ |

review, with the exception of eight studies (S1, S6, S7, S8, S10, S13, S14, and S17). Among these studies, S1, S13, S14, and S17 lack information regarding RQ5 due to the absence of details about the tools used for developing RNN models. Moreover, studies S6, S7, S8, and S10 do not cover RQ7 due to the lack of a comparative evaluation.

After extracting data from the selected studies, our primary aim during the data synthesis step was to combine the collected information effectively to address our research questions. Given the limited number of studies (only 20), and heterogeneous nature of the majority, the application of meta-analysis is infeasible. Instead, we have opted for a descriptive synthesis analysis approach, as outlined in reference [69], which was also utilized in other relevant studies [6, 14, 15, 17, 19].

## 4. Results

Prior to discussing the main results in line with our research questions, as outlined in Section 3.1, we will begin by summarizing the selected studies that fulfill the exclusion/inclusion criteria and have passed the quality assessment.

### 4.1 Overview of the selected studies

This review have carefully selected 20 studies (displayed in Table 6) concentrating on code clones detection through RNN techniques, spanning from 2015 to 2022. Table 9 shows the distribution of these 20 studies and their publication venues. Out of the 20 research studies, seven were published in journals, while the remaining thirteen were published at international conferences.

We obtained valuable insights from prestigious academic journals like IEEE Transactions on Reliability, International Journal of Machine Learning and Cybernetics, and Expert Systems with Applications. Additionally, we considered significant international conferences, including the international conference on automated software engineering (ASE), Conference on Pattern Recognition, and international conference on software analysis, evolution, and reengineering. Fig 4 shows the distribution of studies spanning from 2015 to 2022. Notably, over 65% of these studies were published in 2020 and beyond, indicating a growing research focus on utilizing RNNs for code clone detection in recent years.

Furthermore, we observed that around 65% of the selected studies were published in international conference proceedings, while the remaining 35% appearing in academic journals, as depicted in Fig 5.

### 4.2 RQ1: What RNN techniques have been employed for code clone detection?

Over the last decade, there has been a significant surge in DL applications within software engineering [14, 15, 70]. The advancement of RNNs, as highlighted by Alex Graves [16],

**Table 9. The selected studies and their distribution across venues.**

| ID | Type | Digital Database | Venue | Year |
|---|---|---|---|---|
| S1 | Conference | IEEE | IWCMC | 2020 |
| S2 | Conference | ACM & IEEE | ICSE | 2019 |
| S3 | Conference | IEEE | DSN | 2021 |
| S4 | Journal | Science Direct & Scopus | Expert Systems with Applications | 2021 |
| S5 | Journal | Wiley | Int. Journal of Intelligent Systems | 2022 |
| S6 | Conference | ACM & IEEE | MSR | 2019 |
| S7 | Conference | IEEE & ACM | ASE | 2016 |
| S8 | Conference | ACM & IEEE | MSR | 2018 |
| 90 | Journal | IEEE | IEEE Transactions on Reliability | 2021 |
| S10 | Conference | IEEE | DSA | 2020 |
| S11 | Conference | IEEE | SANER | 2021 |
| S12 | Journal | ACM | ACM Transactions on Software Engineering | 2020 |
| S13 | Conference | IEEE & Scopus | ACPR | 2017 |
| S14 | Conference | ACM | IJCAI | 2018 |
| S15 | Conference | ACM & IEEE | ASE | 2020 |
| S16 | Conference | IEEE | IWCMC | 2020 |
| S17 | Conference | ACM | IJCAI | 2017 |
| S18 | Journal | Science Direct | Computers & Security | 2021 |
| S19 | Journal | Science Direct | Journal of Systems and Software | 2022 |
| S20 | Journal | Springer Link & Scopus | Int. Journal of ML & Cybernetics | 2021 |

demonstrates their proficiency as sequential learners capable of capturing features and long-term dependencies. RNNs techniques are used in a wide array of fields, including research in software cloning. Our review identified the use of nine RNN techniques for detecting code

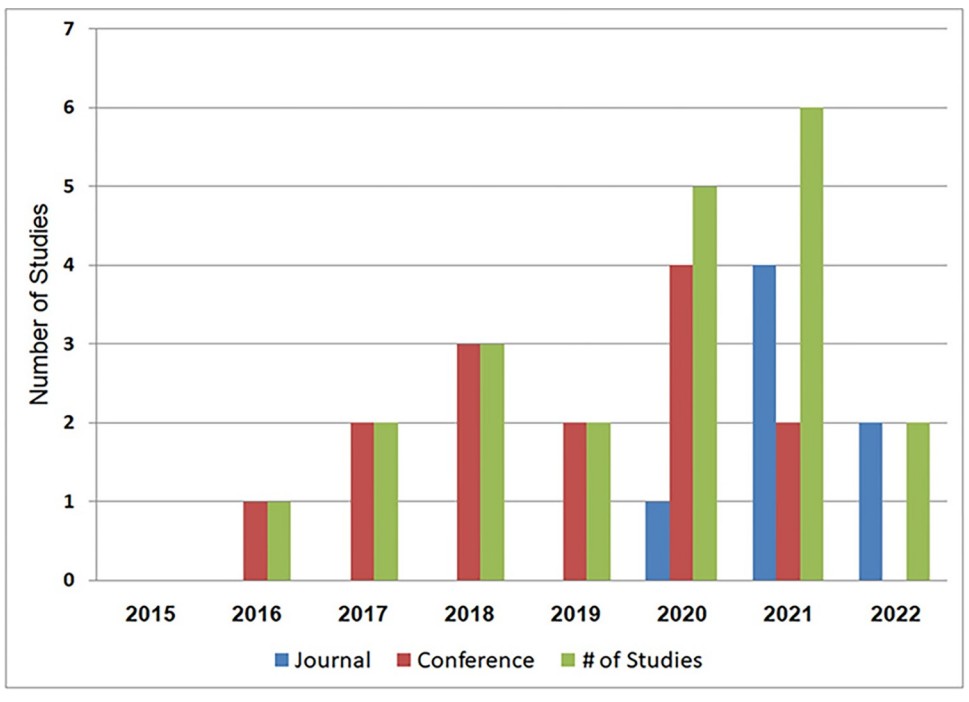

**Fig 4. Number of selected studies based on publication years.**

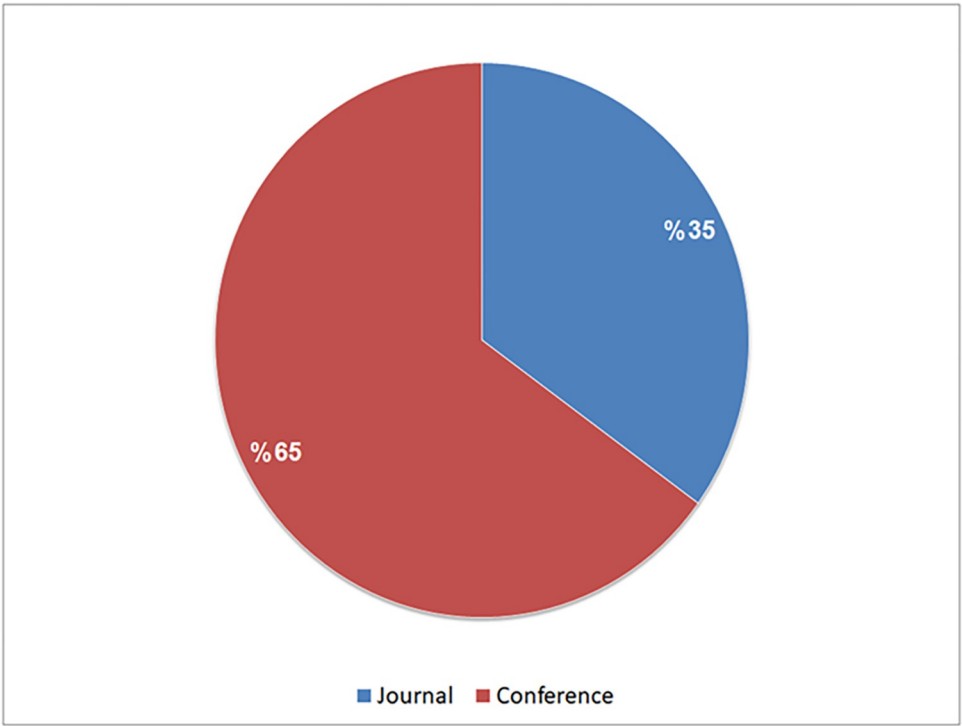

**Fig 5. The distribution of selected studies across different venues.**

clones. Among the selected studies, twelve utilized a single RNN technique, while the rest combined RNN with other DL techniques like CNN, RvNN, or GCNN. Study S9 used two RNN techniques alongside GCNN. With the exception of Bi-RNN, Graph-LSTM, Original GRU, and Bi-GRU techniques, most RNN techniques were appeared in multiple studies. For example studies S1, S6, and S16 utilized the same RNN techniques, Bi-LSTM, while studies S12 and S14 used identical techniques, specifically Child-Sum Tree-LSTM. Table 10 shows the RNN technique used in the selected studies, and Fig 6 shows the frequency of their usage across the selected studies.

Among the RNN techniques used in the 20 studies, the LSTM category was the most frequently used, appearing in 14 of them (70%). It was followed by the TRNN category, which was used in 4 studies (20%). In contrast, the GRU category was used less frequently, appearing in only 2 of the 20 studies (10%). Fig 7 summarizes the RNN categories used across the selected studies.

As observed, Fig 8 indicates a recent trend toward the adoption of RNN techniques in code clone research. The application of RNN in this context emerged in 2016. From 2017 to 2022, researchers predominantly focused on LSTM techniques, reaching their peak usage in 2020 and 2021, being used in 8 out of 11 studies. On the other hand, GRU techniques were used in only two studies, one in 2019 and another in 2020.

### 4.3 RQ 2: What types of clones are most frequently detected using RNN techniques?

To address this research question, we conducted an analysis of the clone types identified in the selected studies, as well as the corresponding performance scores obtained for each clone type.

**Table 10. Distribution of RNN techniques used in the selected studies.**

| Studies | RNN Category | | | | | | | | | # Used RNN Techniques |
|---|---|---|---|---|---|---|---|---|---|---|
| | TRNN | | LSTM | | | | | GRU | | |
| | Original RNN | Bi-RNN | OriginalLSTM | Bi-LSTM | Binary Tree-LSTM | Child-Sum Tree-LSTM | Graph-LSTM | Original GRU | Bi-GRU | |
| S1 | | | | ✓ | | | | | | 1 |
| S2 | | | | | | | | | ✓ | 1 |
| S3 | | | | | ✓ | | | | | 1 |
| S4[+] | | | ✓ | | | | | | | 1 |
| S5[+] | | | ✓ | | | | | | | 1 |
| S6 | | | | ✓ | | | | | | 1 |
| S7[+] | ✓ | | | | | | | | | 1 |
| S8[+] | ✓ | | | | | | | | | 1 |
| S9[+*] | | | ✓ | | ✓ | | | | | 2 |
| S10[+] | | ✓ | | | | | | | | 1 |
| S11[+] | | | | | | ✓ | | | | 1 |
| S12 | | | | | | ✓ | | | | 1 |
| S13 | | | ✓ | | | | | | | 1 |
| S14 | | | | | | ✓ | | | | 1 |
| S15 | | | | | | | | ✓ | | 1 |
| S16 | | | | ✓ | | | | | | 1 |
| S17 | | | | | ✓ | | | | | 1 |
| S18[+] | | | | ✓ | | | | | | 1 |
| S19 | | | | | | | ✓ | | | 1 |
| S20 | ✓ | | | | | | | | | 1 |

RNN: Recurrent Neural Network; Bi-RNN: Bidirectional Recurrent Neural Network

LSTM: Long Short Term Memory; Bi-LSTM: Bidirectional Long Short Term Memory

Tree-LSTM: Tree Long Short Term Memory; Binary Tree-LSTM: Binary Tree Long

Short Term Memory; Child-Sum Tree-LSTM: Child-Sum Tree Long Short Term

Memory; Graph-LSTM: Graph Long Short Term Memory; GRU: Gated

Recurrent Units; Bi-GRU: Bidirectional Gated Recurrent Units

* indicates the study used different RNN techniques

+ indicates the study uses different DL techniques.

Table 11 presents a comprehensive summary of performance scores obtained from the selected studies concerning the detection of different clone types (encompassing Type-I through Type-IV clones), along with the programming language they are associated with.

Based on our analysis presented in Fig 9, the majority of the selected studies (16 out of 20, or 80%) concentrated on the detection of code clones within a single programming language. In contrast, Only four studies (S3, S4, S5, and S6) (20%) delved into the detection of functional clones and semantic clones, across diverse programming languages and platforms. This observation indicates a potential direction for future research. Additionally, it is worth noting that only a small fraction of these studies (10%) of investigated the detection of Type-I to Type-III clones. For instance, White et al. [4], S7, demonstrated the capability of DL in clone detection and study S16 (Zhang et al. [62]), which detected similarities in Scratch code fragments. This suggests that researchers in the remaining studies primarily concentrated on employing RNN techniques to identify the most complex code clones, Type-I and Type-IV.

This underscore the effectiveness of RNNs in detecting the most challenging clones.

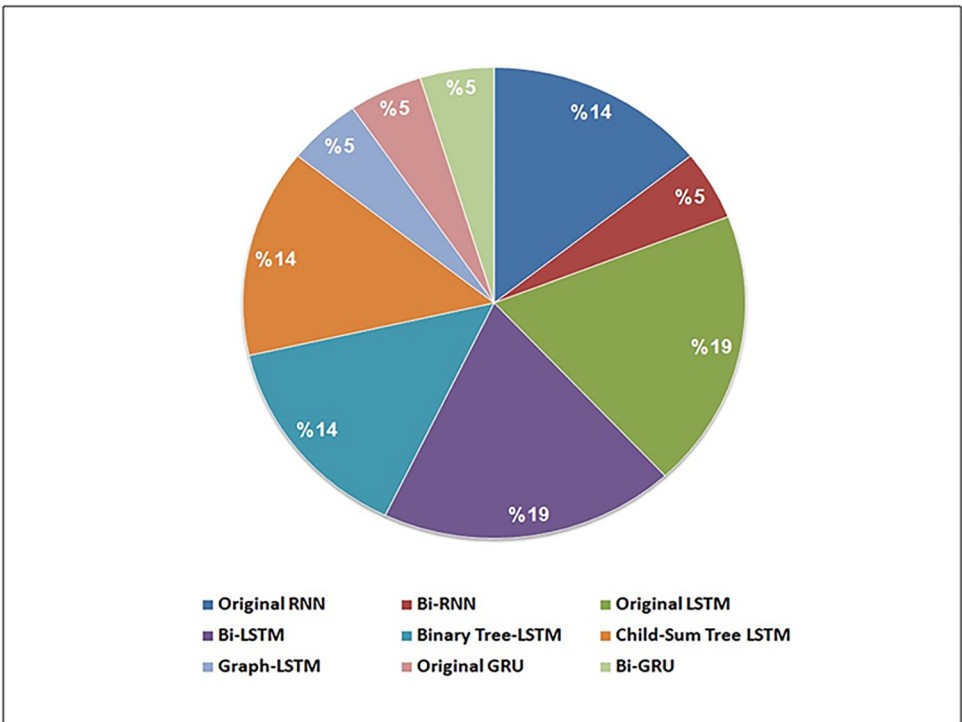

**Fig 6. Analysis of the selected studies in terms of RNN techniques used.**

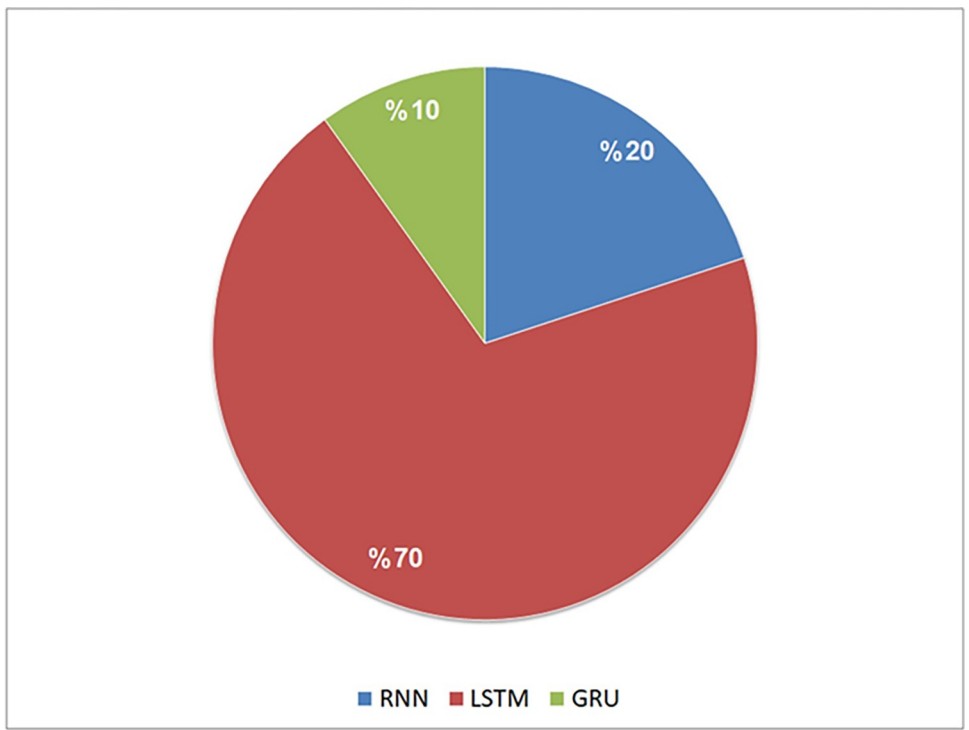

**Fig 7. Analysis of the selected studies in terms of RNN categories used.**

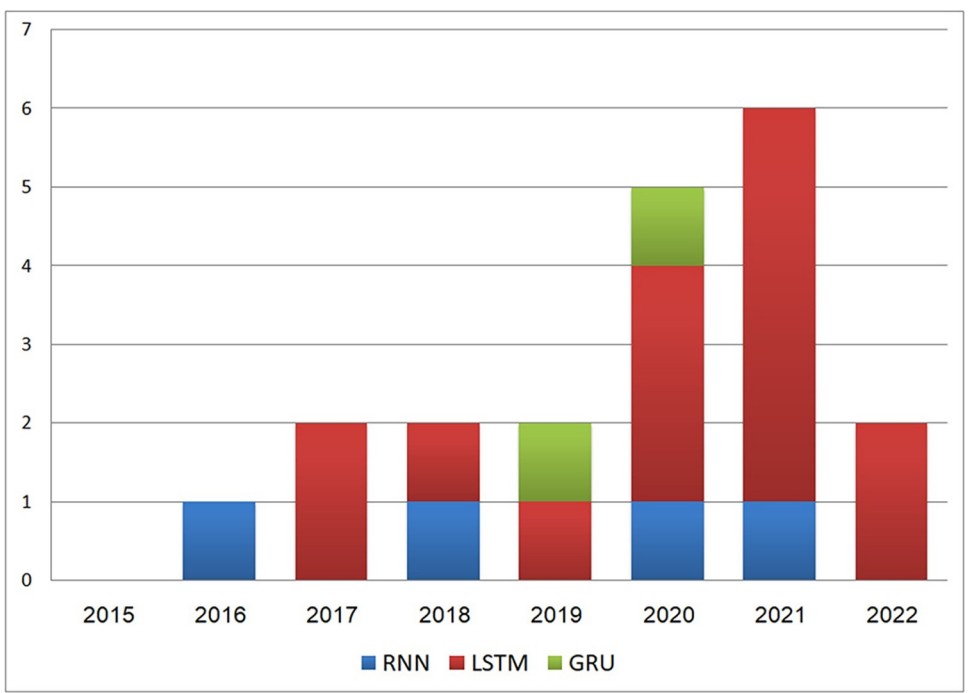

**Fig 8. The distribution of different RNN categories across years.**

**Table 11. The selected studies and the detected clone types and their performance scores in terms of precision, recall, and F-score.**

| Study | PL | CL | Precision | | | | | | Recall | | | | | | F1-score | | | | | | Description |
|---|---|---|---|---|---|---|---|---|---|---|---|---|---|---|---|---|---|---|---|---|---|
| | | | T1 | T2 | VST3 | ST3 | MT3 | WT3/4 | T1 | T2 | VST3 | ST3 | MT3 | WT3/4 | T1 | T2 | VST3 | ST3 | MT3 | WT3/4 | |
| S1 | C | N | - | - | | 94 | | | - | - | | 82 | | | - | - | | 84 | | | Focus on T3&T4 on OJC |
| S2 | Java,C | N | 100 | 100 | - | 99.9 | 99.5 | 99.8 | 100 | 100 | - | 94.2 | 91.7 | 88.3 | 100 | 100 | - | 97 | 95.5 | 93.7 | Focus on all Types BCB |
| S3 | C | Y(CP) | | | | Used AUC, TPR, and FPR to measure the effectiveness of the technique in detecting clones especially Type-IV clones | | | | | | | | | | | | | | | Focus on T4 cross language. |
| S4 | C | Y(CP) | - | - | - | - | - | 96.54 | - | - | - | - | - | 97.25 | - | - | - | - | - | 96.89 | Focus on T4 cross platforms. |
| S5 | C++,Java,C# | Y(CL) | - | - | - | - | - | 84 | - | - | - | - | - | 90 | - | - | - | - | - | 85 | Focus on T4 cross language. |
| S6 | Java,Python | Y(CL) | | | | | | 19 | | | | | | 90 | - | - | - | - | - | 32 | Focus on T4 cross language. |
| S7 | Java | N | | | | Precision values at different levels are reported for 8 real-world Java systems, but clone-type-specific results are not shown | | | | | | | | | | | | | | | Focus on T1&T2&T3 |
| S8 | Java | N | 89 | 82 | 74 | | 67 | | 88 | 84 | 75 | | 18 | | 88 | 83 | 75 | | 29 | | Focus on all clone types |
| S9 | Java | N | - | - | - | - | - | - | - | - | - | - | - | - | 1 | 1 | | 95 | | 98 | Focus on T3&T4 on BCB |
| S10 | Java | N | - | - | | 86 | | | - | - | | 95 | | | - | - | | 91 | | | Focus on T3&T4 |
| S11 | Java | N | - | - | - | - | - | - | - | - | - | - | - | - | - | - | | 92.1 | | | Focus on T3&T4 |
| S12 | C | N | - | - | - | - | - | 88 | - | - | - | - | - | 98 | - | - | - | - | - | 93 | Focus on T4 on OJC |
| S13 | C | N | | | | 84 | | | | | | 94 | | | | | | 88 | | | Focus on T3&T4 |
| S14 | Java | N | | | | 92 | | | | | | 74 | | | | | | 82 | | | Focus on T3&T4 on BCB |
| S14 | C | N | | | | 47 | | | | | | 73 | | | | | | 57 | | | Focus on T3&T4 on OJC |
| S15 | Java | N | | | | 81 | | | | | | 87 | | | | | | 82 | | | Focus on T3&T4 on GCJ |
| S15 | Java | N | | | | 97 | | | | | | 98 | | | 1 | 1 | N | 99 | 99 | 97 | Focus on T3&T4 on BCB |
| S16 | Scratch | N | | | 92 | | | - | | | 98 | | | - | | | 95 | | | - | Focus on T1&T2&T3 |
| S17 | Java | N | | | | 92 | | | | | | 74 | | | 100 | 100 | - | 94 | 88 | 81 | Focus on T3&T4 on BCB |
| S17 | C | N | - | - | - | - | - | 47 | - | - | - | - | - | 73 | - | - | - | - | - | 57 | Focus on T4 on OJC |
| S18 | C | N | - | - | - | - | - | 91.81 | - | - | - | - | - | - | - | - | - | - | - | 91.4 | Focus on T4 |
| S18 | C | N | - | - | - | - | - | 92.51 | - | - | - | - | - | - | - | - | - | - | - | 91.1 | Focus on T4 |
| S19 | Java | N | - | - | - | - | - | 98.5 | - | - | - | - | - | 97.2 | - | - | - | - | - | 97.9 | Focus on T4 on BCB |
| S19 | C | N | - | - | - | - | - | 99.4 | - | - | - | - | - | 99.2 | - | - | - | - | - | 99.4 | Focus on T4 on OJC |
| S20 | Java | N | | | | Used Accuracy metric to evaluate Type-IV clone detection effectiveness. | | | | | | | | | | | | | | | Focus on T4 |

CL: Cross language; PL: Programming Language; CL: Cross Language; CP: Cross Platform; T1: Type-I; T2: Type-II; T3:Type-III; T4: Type-IV; VST3: Very Stronger Type-III; ST3: strong Type-III; MT3: Moderately Type-III; WT3/4: weakly Type-III/Type-IV; AUC: Area under curve; TPR: True Positive Rate; FPR: False positive Rate.

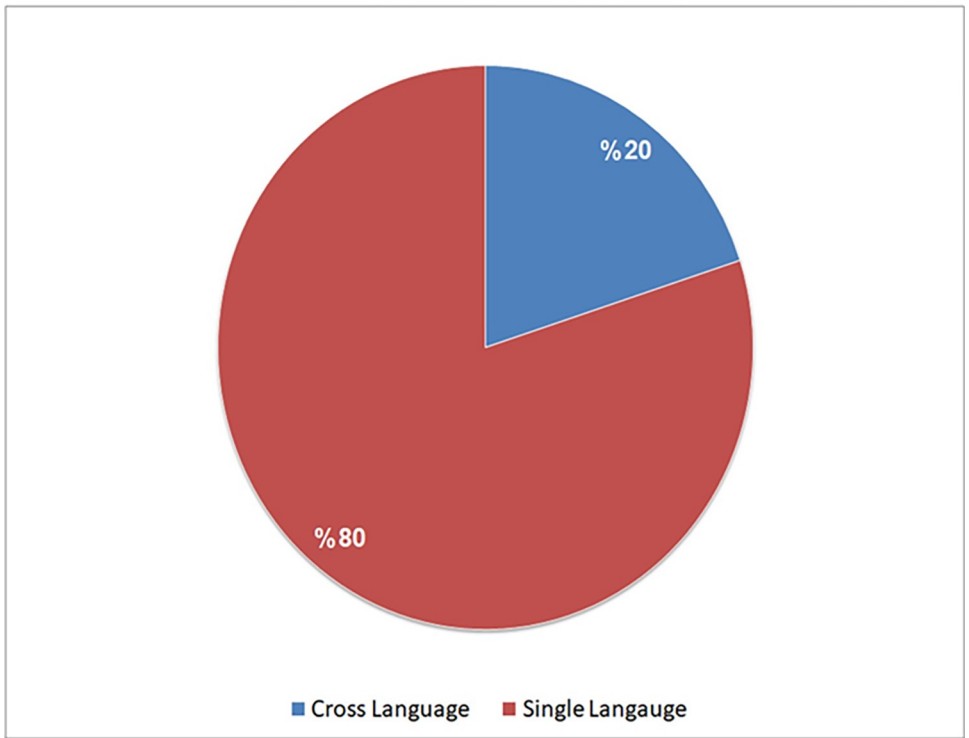

**Fig 9. Analysis of the selected studies in terms of detected clone types.**

Moreover, Fig 10 exclusively displays performance scores for detected Type-III and Type-IV clones, known for their inherent identification complexity, as extracted from these studies. Earlier studies (S6 and S8) exhibited comparatively lower performance in detecting Type-III and Type-IV clones, as evidenced by lower F1-score, recall, and precision. In contrast, some studies chose a more abstract reporting style, such as those observed in studies S3, S7, S13, S14, and S16, where they presented only the final results without specifying the types of detected clones, as listed in Table 11. Conversely, recent research endeavors (S2, S4, S9, S15, S16, S19, and S20) have shown enhanced performance in identifying Type-III and Type-IV clones, including strong Type-III, moderate Type-III, and weak Type-III/Type-IV clones. These improvements were particularly evident in terms of precision, recall, and F1-scores. Among these, S19 demonstrated the highest F1-score performance in the detection of Type-III and Type-IV clones, surpassing its peers in both the BigCloneBench and OJClone benchmarks. This underscores the efficacy of employing Recurrent Neural Networks (RNNs) in the domain of clone detection.

## 4.4 RQ3: What source code representation techniques have been used in RNN applications?

Effective code clone detection relies on code representation techniques, particularly in identifying semantic clones [71]. Different source code representations are used to eliminate uninteresting items and enhance similarity among code fragments. These fragments can be standardized by adjusting their identifiers, layouts, or statements. Alternatively, they can be transformed into intermediate forms like token sequence, abstract syntax tree (AST), or control dependency graph (CDG) [72].

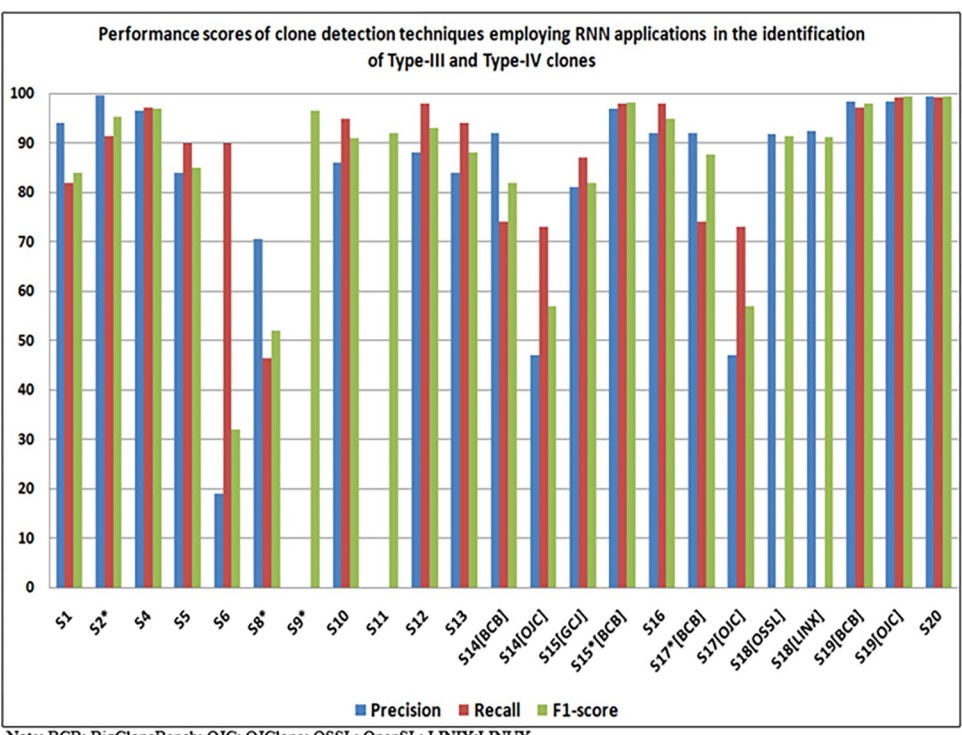

**Fig 10. Analysis the performance scores of the RNN-based clone detection techniques in detecting of Type-III and Type-IV clones.**

To address this question, we analyze source code representation techniques in the selected studies. Table 12 lists the techniques used in these studies. We observed that traditional techniques were predominantly used to represent the source code. These techniques include AST, token sequence, source code text, character sequence, control flow graph (CFG), and program dependency graph (PDG). AST captures the syntactic knowledge of the source code, CFG shows control flow along with its structural information, while PDG covers both control and data flow. Bytecode/assembly instructions illustrate the low-level control flow.

Our findings indicate that the selected studies utilized either individual code representations techniques or combinations of representations techniques to improve clone detection, particularly for clones in the "twilight zone" (Type-III and Type-IV clones) [22]. Among the twenty studies, six (S1, S8, S9, S11, S15, S19) used combined source code representations, while the remaining used individual representations, as detailed in Table 12.

Notably, in Fig 11, we observed that the AST was the most commonly used representation, appearing in 13 studies (46%). Follow that, token sequences were used in 7 studies (25%) and the CFG was used in 5 studies (18%). Other representations were used less frequently.

### 4.5 RQ4: What datasets were used in the selected studies?

This section provides an overview and analysis of the systems and benchmarks used to train and evaluate RNN models. We have compiled a list containing 3 benchmarks, 41 open-source systems, 9 commercial systems, and one student's work, all of which were used in the selected studies. Table 13 presents specific details from this compiled list.

Our observations indicate that in seventeen studies (85%), open-source systems and benchmarks were used for dataset creation. Exceptions include study S3, which used both open-

**Table 12. Source code representations techniques used in the selected studies.**

| Study | Category | Code Representation | | | | | | |
|---|---|---|---|---|---|---|---|---|
| | | AST | Tokens* | Text | Character | CFG | PDG | # count |
| S1 | Bi-LSTM | √ | √ | | | | | 2 |
| S2 | Bi-GRU | √ | | | | | | 1 |
| S3 | Bin Tree-LSTM | √ | | | | | | 1 |
| S4 | Original LSTM | | √ | | | | | 1 |
| S5 | Original LSTM | √ | | | | | | 1 |
| S6 | Bi-LSTM | √ | | | | | | 1 |
| S7 | Original RNN | √ | | | | | | 1 |
| S8 | Original RNN | √ | √ | | | √ | | 3 |
| S9 | Original LSTM | √ | √ | | | √ | | 3 |
| | Bin Tree-LSTM | | | | | | | |
| S10 | Bi-RNN | | | | | √ | | 1 |
| S11 | CS Tree-LSTM | √ | | √ | | | | 2 |
| S12 | CS Tree-LSTM | √ | | | | | | 1 |
| S13 | OriginalLSTM | | | | √ | | | 1 |
| S14 | CS Tree-LSTM | √ | | | | | | 1 |
| S15 | Original GRU | | √ | | | √ | | 2 |
| S16 | Bi-LSTM | | √ | | | | | 1 |
| S17 | Bin Tree-LSTM | √ | | | | | | 1 |
| S18 | Bi-LSTM | | √ | | | | | 1 |
| S19 | Graph-LSTM | √ | | | | | √ | 2 |
| S20 | Original RNN | | | | | √ | | 1 |

*Tokens means a sequences of tokens extracted from the source code by using laxer techniques, sequence of identifiers or literals or constants extracted from AST (leaf node) (e.g., S7),or sequence of bytecode or binary instructions extracted from compiled resources (e.g., S8, S4). Bin Tree-LSTM: Binary Tree-LSTM; CS Tree-LSTM: Child-Sum Tree-LSTM

source and commercial systems, and study S13, which incorporated open-source systems and student work. According to Fig 12, Java systems were the predominant choice, being used in 44% of the studies, followed by C systems, which appeared in 32% of the studies. The remaining systems, coded in languages like C++, C#, Python, and Scratch, were used in 24% of the studies. Notably, certain studies combined systems or benchmarks in different languages to create their datasets; for example, study S2 used both the BigCloneBench (Java) and OJClone (C).

In our findings, We observed that the selected studies used BigCloneBench [73], OJClone [74], and GoogleCodeJam [75] as benchmarks, alongside datasets sourced from code repositories. BigCloneBench is derived from IJDataset 2.0, a repository housing 25,000 open-source Java systems across 43 functionalities, encompassing 8,584,153 true clone pairs and 279,032 false pairs. OJClone, a widely used benchmark, identifies C program clones through solutions submitted by student for 104 unique programming problems. GoogleCodeJam comprises programs from 12 competition problems in diverse languages, hosted by Google.

Furthermore, we observed that four studies (S2, S14, S17, S19) used BigCloneBench and OJClonefor dataset creation, while three studies (S3, S5, S6) produced cross-language datasets. The remaining studies used open-source repositories, as shown in Table 13.

Table 14 lists the publicly accessible datasets from studies, 60% of them being made available.

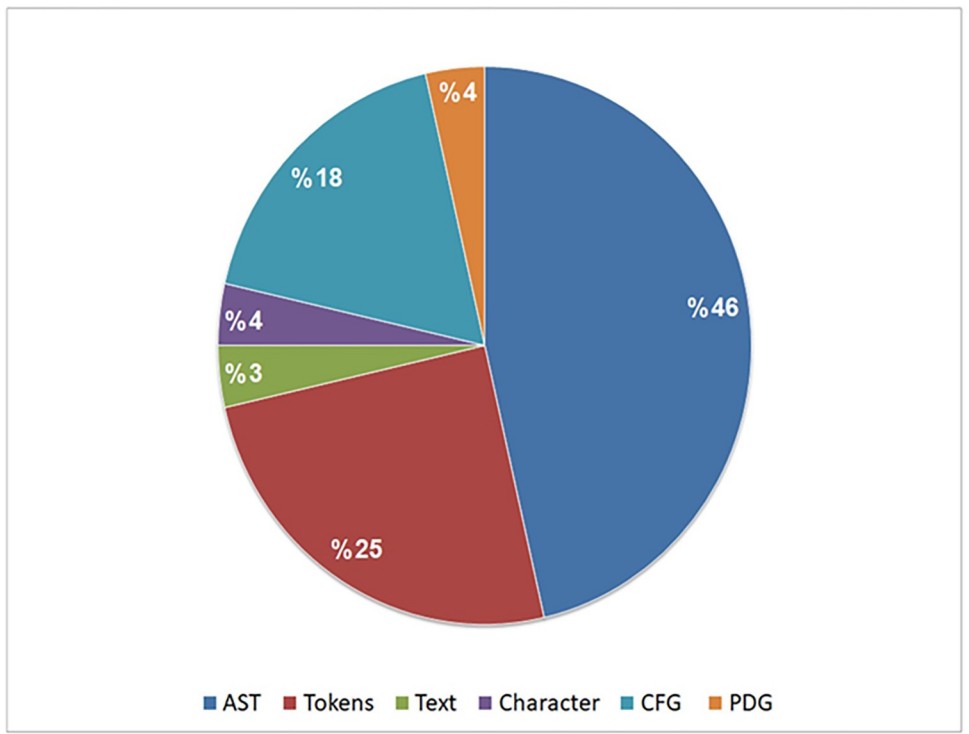

**Fig 11. Analysis of the selected studies in terms of the code representation used.**

**Table 13. Datasets (systems and benchmarks) used in the selected studies to train RNN models.**

| Dataset / Subject System | Type | Language | Studies |
|---|---|---|---|
| OJClone* | Open source | C | S1,S2,S12,S14, S17, S19 |
| BigCloneBench* | Open source | Java | S2,S9,S14,S15, S17, S19 |
| GoogleCodeJam* | Open source | Java | S15 |
| Online judge system on CodingOJ | Open source | C | S1 |
| OpenSSL | Open source | C | S3, S18 |
| Buildroot (busybox, binutils) | Open source | C | S3[@] |
| Firmware (NetGear, Schneider, Dlink, binwalk) | Commercial | - | S3[@] |
| Coreutils, findutils, diffutils, sg3utils, util-linux | Open source | C | S4 |
| Antlr, Argouml, Hibernate, Jhotdraw | Open source | Java | S7, S8 |
| Apache Ant, Carol, Dnsjava | Open source | Java | S7 |
| Ant, Hadoop, Maven, Pmd, Tomcat, Qualitas.Class Corpus, Apache Commons Libraries | Open source | Java | S8 |
| Apache commons imaging, Apache commons math3, Catalano Framework,Colt,Weka(without gui) | Open source | Java | S10 |
| vmarkovtsev | Open source | Java | S11 |
| Linux | Open source | C | S18, S13 |
| ES file downloader, hangman, hangman 2, one cleaner, VPN master | Commercial | Java | S20 |
| GitHub repositories | Open source | C#, Java,C++ | S5[@], S6[@] |
| Apache Software | Open source | Java, Python | S6[@] |
| Student submission | Student | C | S13 |
| Projects from Scratch official website | Open source | Scratch | S16 |

*Indicates the benchmarks, [@]Indicates cross language study

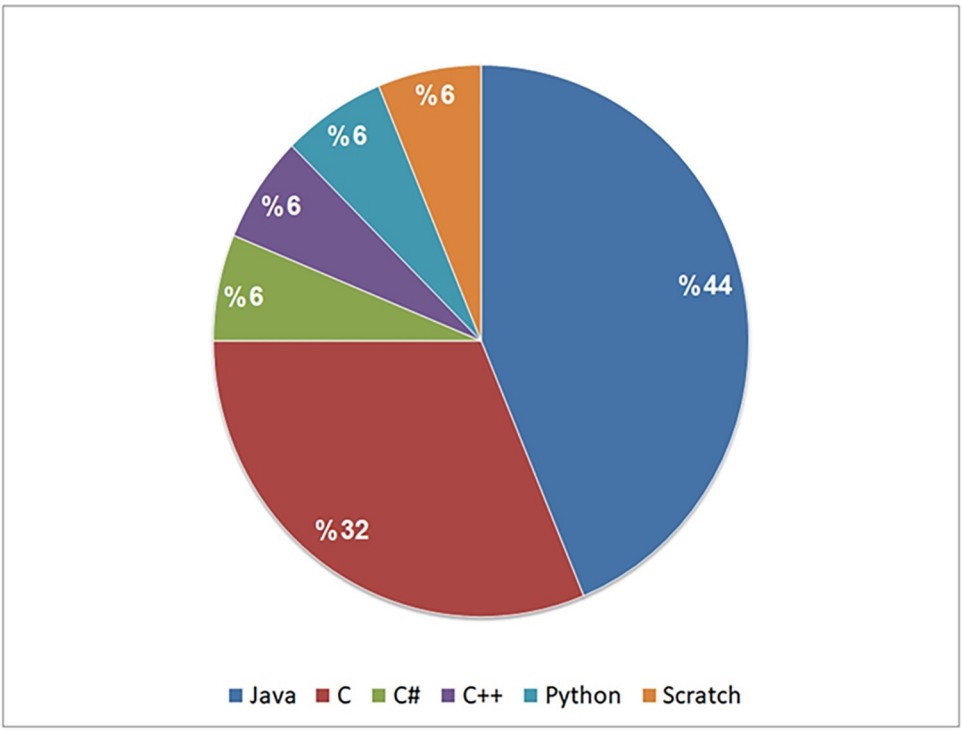

**Fig 12. Analysis of the selected studies in terms of dataset and programming languages used.**

## 4.6 RQ5: What are the most commonly used tools for building RNN models for clone detection?

To address this question, we explored the tools used for implementing RNN models in the selected studies. Our findings reveal the utilization of six tools for implementing these models across these studies. Table 15 presents a comprehensive list of tools, while Fig 13 provides an analysis of the tools used in the selected studies.

Upon analysis, we observed that PyTorch emerged as the most frequently used tool for implementing RNN models, featured in 7 out of 20 studies (35%). TensorFlow closely

**Table 14. The available dataset created by the selected studies.**

| Studies | Links |
|---|---|
| S2 | https://github.com/zhangj1994/astnn |
| S3 | https://github.com/Asteria-BCSD/Asteria |
| S6 | https://www.csg.ci.i.u-tokyo.ac.jp/projects/clone/ |
| S7 | https://sites.google.com/site/deeplearningclone/ |
| S8 | https://github.com/micheletufano/AutoenCODE |
| S9 | https://github.com/preesee/CodeCloneDetection |
| S11 | https://github.com/src-d/datasets/tree/master/Duplicates |
| S14+S17 | https://github.com/clonebench/BigCloneBench<br>http://programming.grids.cn |
| S15 | https://github.com/SCDetector/SCDetector |
| S18 | https://github.com /sunhao123456789/siamese_dataset |
| S19 | https://github.com/YuanJiangGit/Code-Representation-Graph-LSTM |

**Table 15. Tools used in implementing the RNN models.**

| Tool | S₁ | S₂ | S₃ | S₄ | S₅ | S₆ | S₇ | S₈ | S₉ | S₁₀ | S₁₁ | S₁₂ | S₁₃ | S₁₄ | S₁₅ | S₁₆ | S₁₇ | S₁₈ | S₁₉ | S₂₀ |
|---|---|---|---|---|---|---|---|---|---|---|---|---|---|---|---|---|---|---|---|---|
| Pytorch [76, 77] |  | √ | √ |  |  |  |  |  | √ |  | √ | √ |  |  | √ |  |  |  | √ |  |
| Keras [78] |  |  |  | √ |  | √ |  |  |  |  |  |  |  |  |  | √ |  |  |  | √ |
| TensorFlow [79] |  |  |  | √ | √ | √ |  |  |  | √ |  |  |  |  |  | √ |  | √ |  |  |
| RNNLM [80] |  |  |  |  |  |  | √ |  |  |  |  |  |  |  |  |  |  |  |  |  |
| Matlab |  |  |  |  |  |  |  | √ |  |  |  |  |  |  |  |  |  |  |  |  |
| Weka [81] |  |  |  |  |  |  |  |  |  |  | √ |  |  |  |  |  |  |  |  |  |
| Not mentioned | √ |  |  |  |  |  |  |  |  |  |  |  | √ | √ |  |  | √ |  |  |  |

followed, appearing in 6 studies (30%), while Keras was used in 4 studies (20%). Additional tools like RNNLM Toolkit, Matlab, and Weka, each found use in singular studies (S7, S8, and S11, respectively).

Notably, certain studies (S1, S13, S14, and S17) omitted the specification of tools for their implementations, while others (S4, S6, and S16) used both Keras and TensorFlow and study S11 used Pytorch alongside Weka.

### 4.7 RQ6: What evaluation metrics are used to assess the effectiveness of RNN models?

To explore this question, we investigated the evaluation metrics used in the selected studies to measure the effectiveness of RNN models in detecting code clones.

In this review, various evaluation metrics were used in the selected studies to measure RNN model performance in clone detection. Table 16 presents these studies and their corresponding

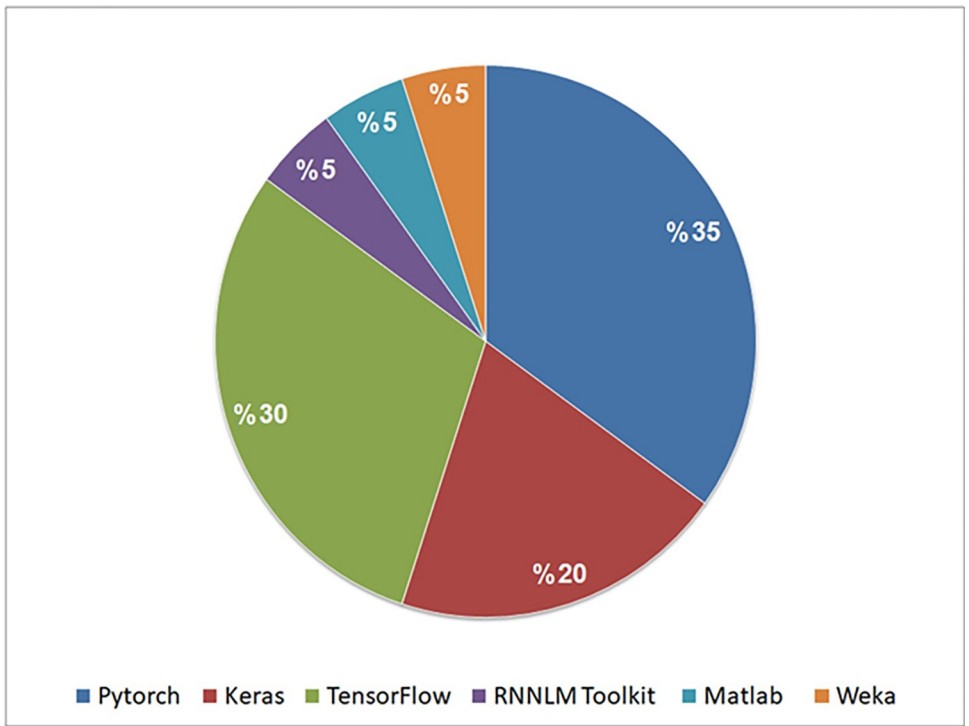

**Fig 13. Analysis of the selected studies in terms of tools used for implementing RNN models.**

**Table 16. Performance evaluation metrics used in the selected studies.**

| Metrics | S₁ | S₂ | S₃ | S₄ | S₅ | S₆ | S₇ | S₈ | S₉ | S₁₀ | S₁₁ | S₁₂ | S₁₃ | S₁₄ | S₁₅ | S₁₆ | S₁₇ | S₁₈ | S₁₉ | S₂₀ |
|---|---|---|---|---|---|---|---|---|---|---|---|---|---|---|---|---|---|---|---|---|
| Recall | √ | √ | | √ | √ | √ | | √ | √ | √ | | √ | √ | √ | √ | √ | √ | | √ | √ |
| Accuracy | | | | √ | √ | | | | | | √ | | | | | | | √ | √ | √ |
| Precision | √ | √ | | √ | √ | √ | √ | √ | √ | √ | | √ | √ | √ | √ | √ | √ | √ | √ | √ |
| F- Score | √ | √ | | √ | √ | √ | | √ | √ | √ | √ | √ | √ | √ | √ | √ | √ | √ | √ | √ |
| Sensitivity | | | √ | | | | | | | | | | | | | | | | | |
| AUC-ROC | | | √ | | | | | | | | | √ | | | | | | √ | | |
| FPR | | | √ | √ | | | | | | | | | | | | | | √ | | |
| FNR | | | | | | | | | | | | | | | | | | √ | | |
| Conf.Mat | | | | | √ | | | | | | | | | | | | | | | √ |

evaluation metrics, while Fig 14 illustrates the utilization of these metrics. Notably, 90% of the selected studies (18 out of 20) used precision and F-score, while 16 studies (80%) used recall. The majority of these studies used a combination of F-score, precision, and recall. Additionally, Six of these studies (S4, S5, S11, S18, S19, S20) integrated accuracy as an evaluation metric. Three studies (S3, S12, and S18) combined AUC-ROC metrics with other evaluation metrics. The remaining metrics were comparatively less commonly used.

## 4.8 RQ7: Which RNN technique yields superior outcomes when evaluated on the same dataset for the same problem?

Comparing and evaluating clone detection techniques presents challenges due to the diversity of subject systems, benchmarks, and the absence of standardized similarity measures [6]. To address this question, we have reported the results of selected studies that applied multiple clone detection techniques to the same datasets or benchmarks and utilized consistent

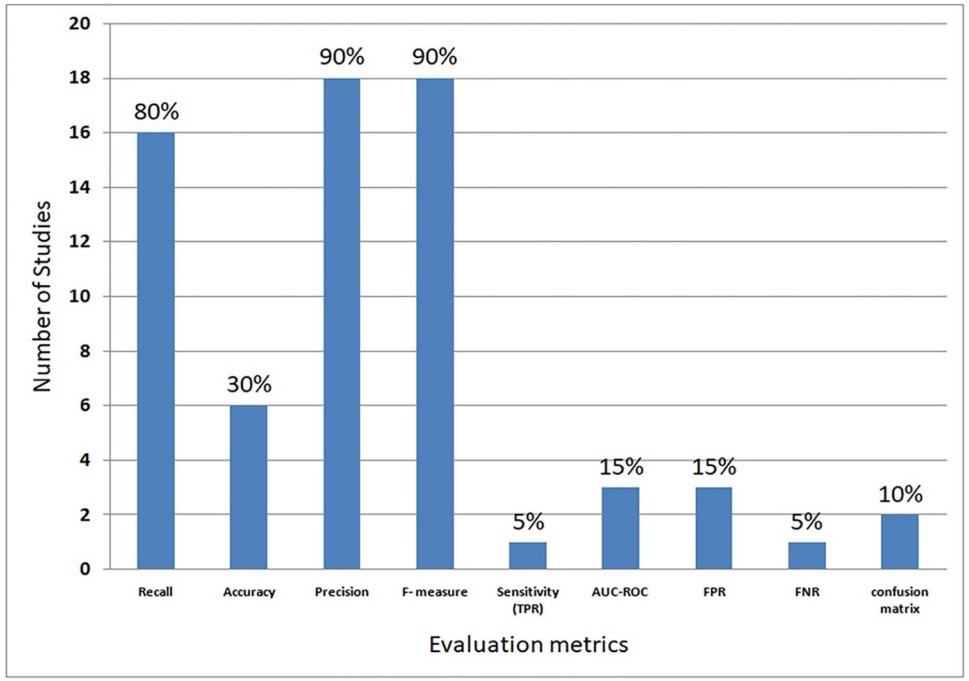

**Fig 14. The performance evaluation metrics in the selected studies.**

evaluation metrics. Our objective is to compare the performance of RNN techniques and determine the top-performing techniques based on precision, recall, and F1-score evaluation metrics.

Table 17 summarizes the performance summarizes the performance results from eight studies (S1, S2, S9, S12, S14, S15, S17, S19) that used similar benchmarks, such as BigClone-Bench, OJClone, and/or Google Code Jam benchmarks. The table provides a details account of

**Table 17. Comparison of RNN techniques to detect clones on BigCloneBench, OJClone,and GCJ.**

| Study | Methods | Outcome | | | | | | Datasets |
|---|---|---|---|---|---|---|---|---|
| S1 | | Precision | Recall | F-Score | | | | OJClone |
| | **PBCS** [48] | **94%** | **82%** | **84%** | | | | |
| | SourcererCC [82] | 44% | 74% | 16% | | | | |
| | Deckard [83] | 98% | 25% | 15% | | | | |
| S2 | | BigCloneBench | | | OJClone | | | BigCloneBenchOJClone |
| | | Precision | Recall | F-Score | Precision | Recall | F-Score | |
| | **ASTNN** [49] | **99.8%** | **88.4%** | **93.8%** | **98.9%** | **92.7%** | **95.5%** | |
| | CDLH [63] | 92% | 74% | 82% | 47% | 73% | 57% | |
| | RAE+ [84] | 76.4% | 59.1% | 66.6% | 52.5% | 68.3% | 59.4% | |
| S9 | | Precision | Recall | F-Score | | | | BigCloneBench |
| | **FCCA** [55] | **98%** | 97% | **98%** | | | | |
| | Deckard [83] | 93% | 2% | 3% | | | | |
| | DLC [4] | 95% | 1% | 1% | | | | |
| | SourcererCC [82] | 88% | 2% | 3% | | | | |
| | CDLH [63] | 92% | 74% | 82% | | | | |
| | TBCNN [21] | 90% | 81% | 85% | | | | |
| | DEEPSIM [85] | 97% | 98% | **98%** | | | | |
| | | Precision | Recall | F-Score | | | | |
| | MTN-a | 84% | 98% | 90% | | | | |
| | MTN-b | 86% | 98% | 91% | | | | |
| | MTN-a w/id | 91% | 98% | 95% | | | | |
| | MTN-B w/id | 91% | 99% | 95% | | | | |
| | Deckard [83] | 60% | 6% | 0.11 | | | | |
| S12 | DLC [4] | 0.7 | 18% | 30% | | | | OJClone |
| | SourcererCC [63] | 97% | 10% | 18% | | | | |
| | CDLH [63] | 21% | 97% | 34% | | | | |
| | CNN [86] | 29% | 43% | 34% | | | | |
| | LSTM | 19% | 95% | 31% | | | | |
| | Bi-LSTM | 18% | 97% | 32% | | | | |
| | Code-RNN [87] | 26% | 97% | 41% | | | | |
| | GGNN [88] | 20% | 98% | 33% | | | | |
| | Tree-LSTM [28] | 27% | 100% | 43% | | | | |
| S14 | | BigCloneBench | | | OJClone | | | BigCloneBenchOJClone |
| | | Precision | Recall | F-Score | Precision | Recall | F-Score | |
| | **CDPU** [60] | 52% | 50% | **51%** | 19% | 17% | **18%** | |
| | Deckard [83] | 93% | 2% | 3% | 99% | 5% | 10% | |
| | DLC [4] | 95% | 1% | 1% | 71% | 0% | 0% | |
| | SourcererCC [63] | 88% | 2% | 3% | 7% | 74% | 14% | |
| | C-DH [89] | 70% | 1% | 1% | 9% | 23% | 13% | |
| | CDLH [63] | 81% | 21% | 33% | 12% | 18% | 15% | |

*(Continued)*

**Table 17.** (Continued)

| Study | Methods | Outcome | | | | | | Datasets |
|---|---|---|---|---|---|---|---|---|
| **S15** | | **BigCloneBench** | | | **Google Code Jam** | | | GJC BigCloneBench |
| | | Precision | Recall | F-Score | Precision | Recall | F-Score | |
| | **SCDetector** | **97%** | 98% | **98%** | **81%** | 87% | **82%** | |
| | SourcererCC | 7% | 98% | 14% | 43% | 11% | 17% | |
| | Deckard | 6% | 93% | 12% | 45% | 44% | 44% | |
| | DLC [4] | 1% | 95% | 1% | 20% | **90%** | 33% | |
| | ASTNN [49] | 94% | 92% | 93% | - | - | - | |
| | | **BigCloneBench** | | | **OJClone** | | | |
| | | Precision | Recall | F-Score | Precision | Recall | F-Score | |
| | CDLH [63] | 92% | 74% | 82% | 47% | 73% | 57% | BigCloneBench |
| **S17** | Deckard [83] | 93% | 2% | 3% | 99% | 5% | 10% | OJClone |
| | DLC [4] | 95% | 1% | 1% | 71% | 0% | 0% | |
| | SourcererCC [63] | 88% | 2% | 1% | 7% | 74% | 14% | |
| | | **BigCloneBench** | | | **OJClone** | | | |
| | | Precision | Recall | F-Score | Precision | Recall | F-Score | |
| | DFS | 98.5% | 97.2% | 97.9% | 99.4% | 99.5% | 99.4% | |
| | SourcererCC [63] | 88% | 2% | 3% | 7% | 74% | 14% | |
| | Deckard [83] | 93% | 2% | 3% | 99% | 5% | 10% | |
| | DLC [4] | 95% | 1% | 1% | 71% | 0% | 0% | BigCloneBench |
| **S19** | CDLH [63] | 92% | 74% | 82% | 47% | 73% | 57% | OJClone |
| | FCCD [90] | - | - | - | 97% | 95% | 96% | |
| | ASTNN [49] | 92% | 94% | 93% | 99% | 93% | 96% | |
| | CodeBert [91] | 94.7% | 93.4% | 94.1% | 99.9% | 96.1% | 97.9% | |
| | GraphCodBert [92] | 94.8% | 95.2% | 95% | 100% | 97.7% | 98.8% | |

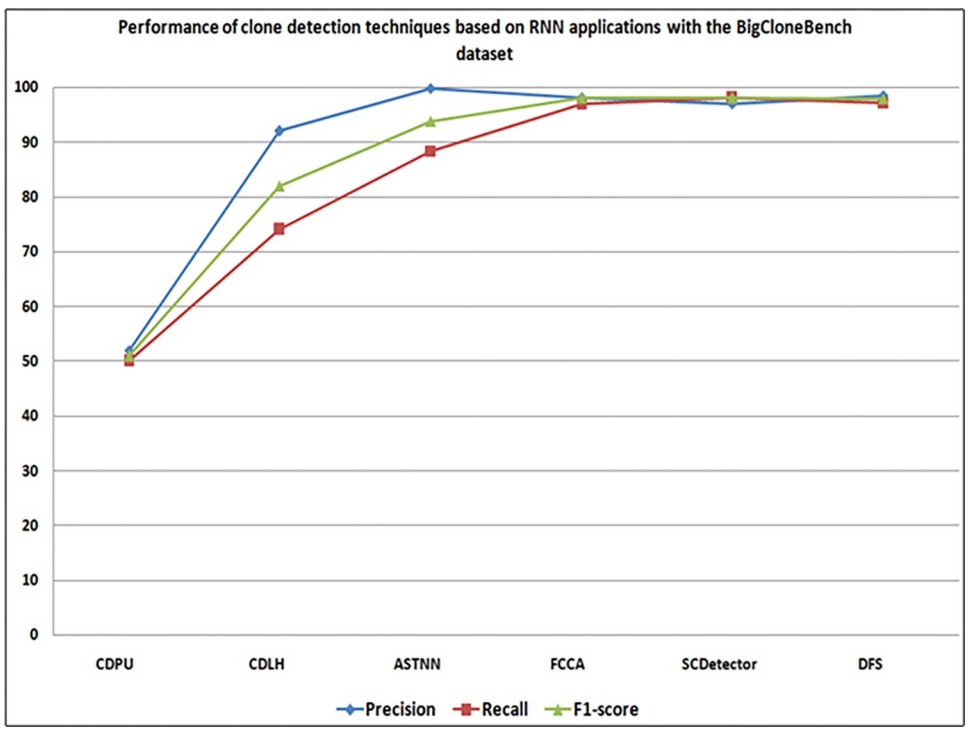

**Fig 15. Comparison of detection techniques based on RNN applications with BigCloneBench dataset.**

the code clones detection techniques evaluated across these studies, encompassing key evaluation metrics (precision, recall, and F1-Score), as well as the subject systems under investigation. Furthermore, in Table 17, we can find studies in which an RNN-based technique was compared against other clone detection techniques. Dong et al. [48] (S1) proposed the PBCS technique, using Bi-LSTM to identify semantic similarity in students' program assignments. When compared to SourcererCC [82] and Deckard [83] using the OJClone dataset, the results demonstrated PBCS's superior performance. Jian et al. [49] (S2) proposed AST-based neural network (ASTNN) to represent source code. This technique involves decomposing large abstract syntax trees (ASTs) of code fragments into a sequence of smaller statement trees, upon which they performed tree-based neural embeddings. In their work, they conducted a comparative analysis of this approach with CDLH [63] and RAE+ [84], revealing that ASTNN demonstrated superior performance when compared to other ML-based techniques for clone detection. These results were consistent across the OJClone and BigCloneBench datasets. Wei et al. [55] (S9) introduced the FCCA technique, a functional code clone detection technique that leverages deep learning (LSTM, Tree-LSTM, and GCNN) and integrates diverse code features, encompassing unstructured elements like token sequences and structured components such as Abstract AST and CFG. The incorporation of an attention mechanism enhance detection accuracy by focusing on critical code segments and features. In a comparative study against other techniques, including Deckard [83], DLC, SourcererCC [82], CDLH [63], TBCNN [21], DEEPSIM [85], FCCA emerged as the superior performer in terms of recall, precision, and F1-score on the BigCloneBench dataset. DEEPSIM achieved a similar F1-score to FCCA but with lower precision and higher recall. Wenhan et al. [58] (S12) proposed a novel techniques for source code representation, referred to as the modular tree network (MTN). Unlike previous neural network models based on tree structures, the modular tree network refines the child-sum tree-LSTM's basic structure and can effectively capture semantic distinctions among various types of AST substructures. They conducted a comprehensive comparison with existing techniques, including Deckard [83], DLC [4], SourcererCC [82], CDLH [63], CNN [86], LSTM, Bi-LSTM, Code-RNN [87], GGNN [89], and Tree-LSTM [28]. The results demonstrate that the proposed technique outperforms other methods in terms of F1-score, particularly on the OJClone dataset. Wei and Ming [60] (S14) introduced CDPU for identifying functional clones using Siamese GRU. CDPU leverages adversarial training to distinguish non-clone pairs that appear similar but behave differently. In comparisons with Deckard [83], DLC [4], SourcererCC [82], CDLH [63], and C-DH [89] on the BigCloneBench and OJClone benchmark, CDPU CDPU outperforms in F1-score and recall on both benchmarks but exhibits lower precision on BigCloneBench and OJClone. Yueming et al. [61] (S15) proposed SCDetector, a novel method for detecting functional code clones that combines token-based and graph-based techniques. They employed a siamese architecture neural network with GRU to create a more accurate and efficient model. SCDetector was compared to other techniques, including ASTNN [49], Deckard [83], DLC [4], and SourcererCC [82], on the Google Code Jam and BigCloneBench benchmarks. Results showed that SCDetector outperformed other methods on the BigCloneBench benchmark in F1-score, precision, and recall, and also performed well on the Google Code Jam benchmark, despite having the second-highest recall. Wei and Li [63] (S17) presented CDLH, a clone detection approach using tree-based LSTM to identify semantic clones. CDLH was compared to Deckard [83], DLC [4], and SourcererCC [82] on the BigCloneBench and OJClone benchmarks, with results indicating that CDLH's superior performance in terms of F1-score on both benchmarks. Ullah et al. [31] (S19) proposed a technique for source code representation using a semantic graph to capture both syntax and semantic features. Their study compared this technique to several others, including Deckard [83], DLC [4], SourcererCC [82], CDLH [63], FCCD [90], ASTNN [49], CodeBert

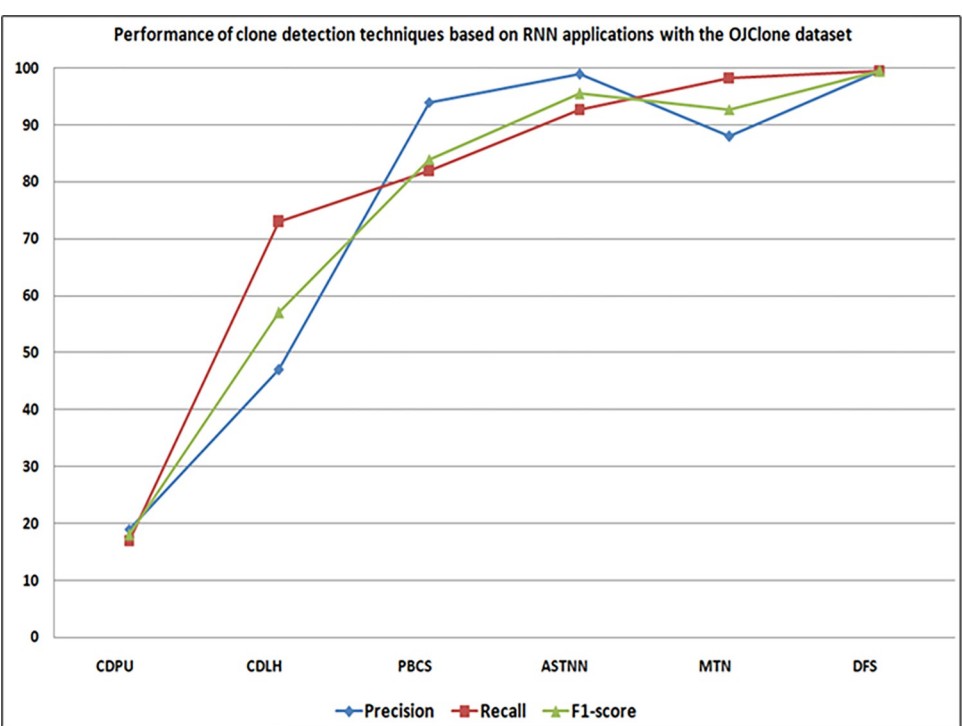

**Fig 16. Comparison of detection techniques based on RNN applications with OJClone dataset.**

[91], and GraphCodeBert [92]. The results indicate that their technique outperforms other methods on the BigcloneBench and OJClone datasets in terms of F1-score and recall.

Figs 15 and 16 present a comparative analysis of studies employing RNN applications for code clone detection on the BigCloneBench, and OJClone datasets. ASTNN [49] (S2) exhibited exceptional precision, while SCDetector [61] (S15) achieved higher recall and F1-score when compared to other clone detection approaches on the BigCloneBench dataset. On the other hand, Ullah et al.'s DFS technique [31] (S19) showed strong performance on OJClone, excelling in terms of precision, recall, and F1-score, closely followed by ASTNN [47]. Additionally, the DFS technique [31] (S19) outperformed other techniques on both benchmarks in terms of precision, recall, and F1-score, with ASTNN [47] as the next best performer. Conversely, CDPU [60] (S14) and PBCS [48] (S1) exhibited the lowest performance in precision, recall, and F1-score on both benchmarks when compared to other techniques.

The mean F1-Score values presented in Table 18 and depicted in Fig 17 offers valuable insights into the progress achieved in RNN techniques for clone detection.

**Table 18. Average of F1-score across years.**

| Year | F1-Score | Average |
|------|----------|---------|
| 2016 | -(S7) | - |
| 2017 | 89 (S13), 82 (S17 on BCB), 57 (S17 on OJC) | 76 |
| 2018 | 96 (S8), 85 (S8), 51 (S14 on BCB), 18 (S14 on OJC) | 62.5 |
| 2019 | 93.8 (S2 on BCB), 95.5 (S2 on OJC), 32 (S6) | 73.77 |
| 2020 | 84 (S1 on OJC), 91 (S10), 92.75 (S12 on OJC), 98 (S15 on BCB), 82 (S15 on GCJ), 95 (S16) | 90.46 |
| 2021 | -(S3), 97.97 (S4), 98 (S9 on BCB), 92.1 (S11), 91.4 (S18 on Openssl), 91.1 (S18 on Linux), 96.2 (S20) | 94.41 |
| 2022 | 85 (S5), 97.9 (S19 on BCB), 99.4 (S19 on JOC) | 94.10 |

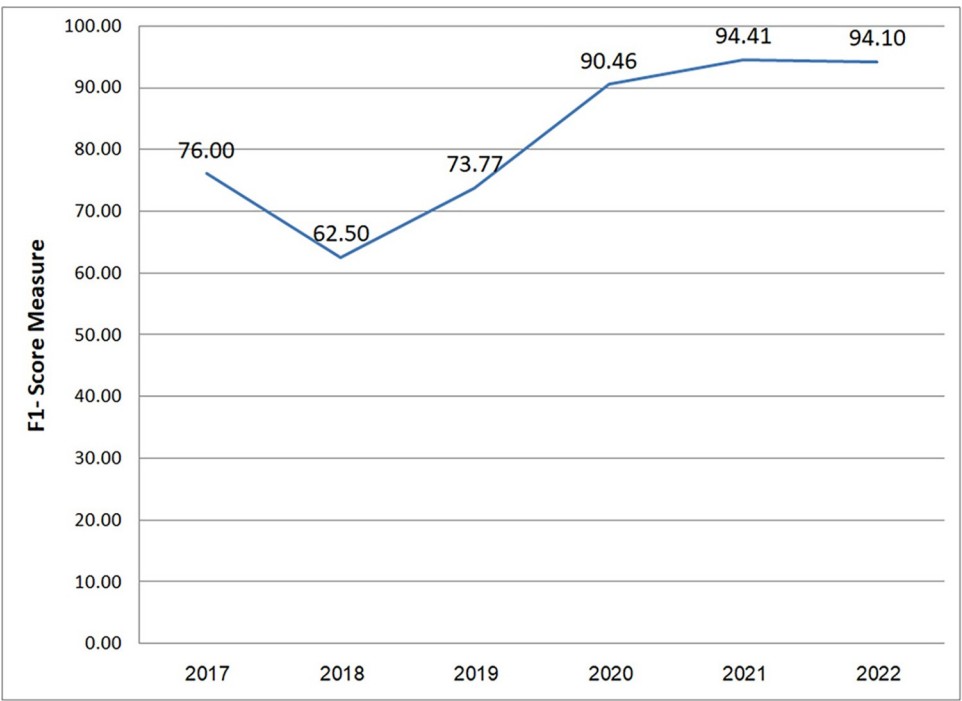

**Fig 17. Average F-score across years.**

Fig 17 demonstrates a notable improvement in performance over time, indicating the effectiveness of these techniques. The relatively lower F-Score values observed in 2017 and 2018 can be attributed to the early stage of development for deep learning techniques in this domain.

Tables 17 reveals the significance of the works by Wei and Ming [63] (S17), White et al. [4] (S7), and Hua et al. [55] (S9) in clone detection, as they were frequently benchmarked against other techniques. However, four studies (S6, S7, S8, S10) did not include comparative evaluations. Furthermore, the selected studies compared their techniques with traditional methods, with Deckard and SourcererCC being the most commonly utilized traditional techniques of comparison.

We have observed that eleven studies (S1, S3, S4, S6, S9, S12, S13, S15, S16, S17, S18) employed a Siamese architecture to improve the performance of RNN model. This architecture has gained popularity in facial recognition tasks and various NLP applications, including sentence similarity assessment [93, 94].

In the context of validating techniques used in the selected studies, two validation configurations were used. Table 19 presents these techniques and their usage in the selected studies. The findings indicate that 13 studies (65%) applied percentage-based split validation, while 4 studies (20%) opted for K-fold cross-validation. However, in three studies (S7, S14, S17) (15%), the specific validation techniques used were not specified.

**Table 19. Validation of learning techniques in the selected studies.**

| Validation Technique | Studies |
| --- | --- |
| K-fold cross-validation | S8, S9, S13, S18 |
| Percentage-Based Split | S1,S2,S3,S4,S5,S6, S10,S11, S12, S15, S16, S19, S20 |
| Not mentions | S7,S14, S17 |

The choice of validation techniques, whether through the application of percentage-based training and testing sets or the adoption of k-fold cross-validation, depends on various factors, including the dataset size used for training and testing machine learning models.

Deep learning techniques require large datasets for effective training and achieving optimal results [95]. Notably, the majority of the selected studies (65%) prefer percentage-based data splitting, randomly dividing the dataset into training, validation, and testing sets, due to their substantial dataset size and the simplicity of this technique. However, it is important to recognize that the inherent randomness in creating these subsets can significantly impact the predictive model's performance [96].

In the K-fold cross-validation, the dataset is randomly partitioned into k equally sized subsets or folds. From these k subsets, one is designated as a test set, while the remaining k-1 subsets are used for training the model. This process is then repeated k times, ensuring that each of the k subsets is employed as the test set exactly once. However, it is important to note that running K-fold cross-validation multiple times increases the computational cost and time, as it necessitates training and testing the model K times. This can become particularly resource-intensive for large datasets or complex models [97]. Consequently, some studies opt for alternative validation techniques, which may explain why only four studies in this context have chosen to use the K-fold cross-validation technique.

## 5. Discussion

In this paper, we investigate various studies to determine the extent of research conducted in the field of code clones using RNN techniques. This section presents the main findings of our work, based on the research questions we posed, along with recommendations for future research that may be of interest to the research community.

RQ1: We conducted an extensive search across various sources, including ScienceDirect Digital Library, Scopus Digital Library, and IEEE Xplore Digital Library, and identified 20 studies that utilized RNN techniques for detecting code clones. for further details, please refer to Table 6 and Fig 4.

Our findings reveal that nine RNN techniques were used to identify code clones, categorized as TRNN, LSTM, and GRU. Among these, the LSTM category was the most frequently used (70%), followed by TRNN (20%) and GRU (10%). Additional details can be found in Table 10 and Figs 6 and 7.

Among the 20 reviewed studies, only S9 utilized multiple RNN techniques, specifically OLSTM and Binary Tree-LSTM. None of the studies integrated RNN techniques from different categories, highlighting the limited utilization of diverse RNN techniques for code clone detection. Consequently, further research is required to explore the efficacy of combining different RNN techniques for this purpose.

Traditional RNNs encounter challenges with gradients that hinder their ability to learn long-term dependency [98, 99]. To address this issue, Hochreiter and Schmidhuber [26] devised LSTM, which incorporates a memory cell for extended state retention. However, the use of LSTM can lead to training delays [60, 100]. To mitigate this, Bouaziz et al. [101] proposed a solution in the form of parallel LSTM, enabling concurrent stream processing. Furthermore, the sentence-level LSTM [102] enhances LSTM performance in NLP tasks by facilitating parallel operations and improving the extraction of complex information from text. Interestingly, despite the demonstrated superior performance of parallel LSTM and sentence-level LSTMs in NLP tasks [103, 104], none of the reviewed studies employed parallel LSTM or sentence-level LSTM for code clone detection. Therefore, further research is needed to investigate the applicability of these LSTM techniques to code clone detection.

Furthermore, Marcin et al. [105] introduce a less complex Tree-Structured Gated Recurrent Unit (GRU) architecture compared to the prominent Tree-Structure LSTM model used in sentiment analysis. Despite this, no prior exploration has been conducted on the potential of the Tree-Structure GRU's for detecting code clones. Consequently, additional research is necessary to investigate its capabilities in identifying code clones.

RQ2: Code clones are categorized as syntactic (Type-I, Type-II, and Type-III), and semantic (Type-IV) [7]. Our findings indicate that most of the selected studies concentrated on identifying semantic clones within code written in a single programming language. Only four studies (S3, S4, S5, and S6) examined detection across different languages, or platforms. Therefore, additional research is required to investigate how RNN techniques can identify semantic clones across different programming languages and platforms. Additionally, our findings indicate that only two studies (S3, S4) used RNN techniques to detect semantic similarities among binary code fragments. This highlights the need for further research in this vital area, given its significance in security applications like vulnerability discovery and malware analysis [50].

RQ3: The performance of DL models is notably influenced by the selected code representation [71]. Specifically, the accuracy of DL models, including RNNs models, heavily relies on the specific code representation used [71, 72]. Table 12 shows that the dominant code representation in the selected studies is the AST, while only two studies (S4 and S8) using lower-level representations like assembly code and Bytecode instructions.

Compilation and decompilation techniques normalize code structures by transforming syntactically different but semantically similar structures into a consistent binary form [72, 106, 107]. Therefore, we suggest harnessing RNN techniques to identify semantic clones through lower-level representations, like three-address code, Bytecode/Assembly instructions, or integrate both higher and lower-level code representations.

While existing research extensively examines code clone identification within single programming languages, limited attention has been given to detecting clones across multiple languages. With the increasing prevalence of multi-language software development in major applications like twitter, Uber, and Netflix, the demand for cross-language clone detection is growing. Notably, only two studies (S6 and S5) have utilized RNN techniques for detecting clone across language, using AST representations to capture significant features. Nevertheless, further investigation is needed to effectively leverage RNN techniques for detecting code clones across languages, using different code representation techniques.

It is well-established that effective code clone detection relies heavily on code representation techniques, particularly in the context of identifying semantic clones [71]. Consequently, this observation opens up opportunities for further and more comprehensive research, with a particular emphasis on the thorough examination of code representation techniques in the context of machine learning and deep learning techniques across various research domains.

**RQ4**: Based on our research findings, the majority of selected studies focused on subject systems coded in Java and C programming language. This choice was driven by the availability of benchmarks like BigCloneBench, GoogleCodeJam, and OJClone.

BigCloneBench benchmark, offering a large-sized dataset and support for a wider range of functionalities (43 in total), is superior for Java clone datasets compared to GoogleCodeJam. It provides a significant advantage, as it contains a substantial proportion (98%) of semantic clones. On the other hand, among the selected studies that focused on the C programming language, OJClone stood out as the sole publicly available benchmark. Consequently, there is a need for research to create new benchmarks for programming language like Python, Scratch, and C# to advance the research on detecting clones within these languages.

Additionally, we noted two studies (S5 and S6) that proposed techniques for detecting semantic clones across multiple programming languages, specifically Java, C#, C++, and

Python. These studies created their datasets from GitHub repositories and Apache Software. However, the absence of validated benchmarks hinders the assessment of their effectiveness. Therefore, additional research is necessary to create dependable benchmarks for evaluating studies on cross-language semantic clone.

In recent years, numerous studies have addressed detecting Java code clones using the Big-CloneBench benchmark. Certain Java clone detectors concentrate exclusively on Bytecode-level clone, neglecting Java source code. To address this limitation, Schafer et al. [108] introduced Stubber, a tool that compiles Java source code into Bytecode without dependencies. This advancement allowed Bytecode-based clone detectors to be evaluated on BigCloneBench for more than 95% of Java source files. Therefore, Additional research is necessary to establish benchmarks for evaluating the efficacy of binary code-level clone detection techniques.

Furthermore, We observed that most selected studies used open-source for datasets, with the exception of two studies (S3 and S20) which utilized commercial software. It's important to note that clones could be present within commercial software. Therefore, there is a need to broaden research in this domain to include commercial applications.

**RQ5**: Our analysis showed that among the six tools used to build RNN models in the selected studies (as detailed in Table 15 and Fig 13), Pytorch and TensorFlow were the most common, being used in 55% of the studies. This prevalence can be attributed to their established recognition, open-source nature [76, 77, 79], and widespread adoption for neural networks [109]. Researchers have a promising opportunity to enhance code clone detection by considering alternative frameworks with high rankings for implementing DL models, such as MXNet [110] and Theano [111].

**RQ6**: Our findings highlight that the prevalent evaluation metrics for the selected studies include precision, recall, F1-score, accuracy, and AUC-ROC. However, it's crucial to acknowledge that relying solely on threshold-dependent metrics might introduce bias in interpreting ML model performance [96]. To address this, incorporating threshold-independent metrics like Matthew's Correlation Coefficient (MCC) [112] and/or the Area Under the ROC Curve [113] is recommended. Moreover, it should be noted that evaluating studies with BigClone-Bench may result in different recall values due to incomplete labeling. Furthermore, some studies employ non-universal precision evaluation metrics, requiring manual validation of all clones. Thus, a comprehensive evaluation of existing RNN code clone prediction models is imperative.

**RQ7**: Most of the selected studies commonly use BigCloneBench and OJCloneasas benchmarks for assessing semantic clone detection techniques. These studies consistently exhibit enhanced performance on the BigcloneBench and/or OJClone datasets.

Based on the data in Table 17 and Figs 15 and 16, which provide a comparative analysis of studies employing RNN applications for code clone detection on the BigCloneBench, and OJClone datasets, we clearly observed that ASTNN [49] (S2) demonstrated exceptional precision at 99.8%, while SCDetector [61] (S15) achieved higher recall and F1-score when compared to other clone detection approaches on the BigCloneBench dataset, reaching 98% for both metrics. On the other hand, Ullah et al.'s DFS technique [31] (S19) exhibited strong performance on OJClone, boasting impressive precision, recall, and F1-score values of 99.4%, 99.5%, and 99.4%, respectively, closely followed by ASTNN [47] with precision, recall, and F1-score at 98.9%, 92.7%, and 95.5%. Additionally, the DFS technique [31] (S19) outperformed other techniques on both benchmarks in terms of precision, recall, and F1-score, with ASTNN [47] as the next best performer. Conversely, CDPU [60] (S14) and PBCS [48] (S1) displayed the lowest performance in precision, recall, and F1-score on both benchmarks when compared to other techniques.

Furthermore, we have found that studies (S3, S4, S5, and S6) propose cross-platform semantic similarity detection techniques. Specifically, studies (S5 and S6) introduce RNN-based techniques to identify analogous code fragments across different languages. In contrast, study (S4) focuses on detecting binary code similarity across varied architectures, compilers, or optimizations. Additionally, study (S3) presents an innovative RNN-based technique for detecting binary similarity across platforms. Notably, it is worth noting that cross-language models exhibit relatively poorer performance compared to single-language models [14], highlighting the necessity for further research in this domain.

In the context of validating deep learning techniques used in the selected studies, it is important to recognize that the inherent randomness in dataset creation can significantly impact the performance of predictive models. To address this, we strongly recommend following the guidelines outlined by Hall et al. [96] to ensure the robust design of empirical investigations. Particular attention should be given to the use of threshold-independent metrics and the implementation of preprocessing techniques such as feature selection and data balancing to enhance the setup of deep learning models. Furthermore, our observations highlight the importance of using manually validated datasets that effectively capture code clones within code fragments.

## 6. Threats to validity

This section highlights various potential threats that could impact the validity of this SLR, despite the authors' efforts to implement a rigorous methodology to ensure accurate findings.

A significant threat involves the challenge of identifying all relevant studies, a common issue encountered in systematic literature reviews [20, 114]. There is a possibility that certain studies focusing on RNN techniques in code clone analysis might have been unintentionally missed. To mitigate this threat, we implemented a strategy that entailed the selection of seven widely acknowledged research databases: Science Direct, Springer, ACM, Scopus, IEEE, Willey, and World Scientific. Additionally, we conducted a snowballing process by examining the references of selected studies and incorporating relevant studies.

Another threat revolves around the formulation of an effective and comprehensive set of search terms. To address this concern, we developed a comprehensive search strategy. This strategy involved extracting keywords from the research questions, identifying synonyms or alternative spellings, and confirming their presence in related studies.

To reduce the risk of including irrelevant studies or unintentionally excluding relevant ones during the study selection phase, we implemented a two-step process. Initially, two authors independently applied the inclusion and exclusion criteria to filter out irrelevant studies. In cases of disagreement, discussions among the authors were conducted to resolve any discrepancies.

Potential threats also arise during the phase of quality assessment and data extraction. To mitigate these risks, we executed a formal procedure encompassing the creation of a set of rigorous quality assessment criteria and a comprehensive data extraction form. The assessment of study quality was conducted independently by two authors. After that, third author independently selected a separate set of studies through a randomized process and conducted a double-checked of the selection process, any disagreements among authors were addressed and resolve. Following this, a single author meticulously reviewed each study, extracting and documenting relevant information using a predefine template. In addition, other authors independently examined a randomized selection of studies, engaging in a meticulous cross-verification process to ensure the accuracy of results. Any differences among authors were addressed through to reach at a consensus.

## 7. Conclusion

This study reports findings from a systematic literature review (SLR) that investigates the utilization of RNN techniques for code clone detection. The review involved comprehensive searches across seven online databases to identify relevant studies. Ultimately, twenty studies published between 2015–2022 were selected to address the research objectives. These selected studies were grouped into three categories: Traditional recurrent neural network (TRNN), long short-term memory (LSTM), and gate recurrent unit (GRU). Each category encapsulating distinct techniques. Research questions were formulated to explore the current state of RNN techniques in clone research from various perspectives, including the specific RNN techniques used, types of detected clones, code representation techniques, evaluation metrics, popular datasets, studies achieving better results when compared to alternative approaches, and prevalent tools for implementing RNN models.

The results indicate that among the RNN techniques, the LSTM variations like OLSTM, Bi-LSTM, Binary Tree-LSTM, and Child-Sum Tree-LSTM, are most commonly used. Furthermore, the AST is the prevalent code representation technique, used in 13 studies. Our review also reveals that, researchers predominantly favored datasets and benchmarks written in the Java programming language. However, it's essential to highlight that only 12 studies made their datasets available for further use. Regarding the tools used for implementing RNN models, Pytorch stands as the most commonly used tool, closely followed by TensorFlow. However, regarding the evaluation metrics, most studies used F-score, precision, and recall to assess the performance of the implemented RNN models.

As a result of this study, several observations should be considered for future research using RNN techniques for code clone detection:(1) No studies have investigated the potential of new RNN architectures, such as parallel LSTM, sentence-level LSTM, and Tree-Structured GRU, within the field of code clone research. (2) Only two studies have used RNN techniques to detect semantic clones across diverse programming languages. (3) Only two studies have used RNN techniques to identify semantic similarity among binary code fragments. (4) Only one study has used a combination of PDG and AST to represent code fragments. (5) Manually validated benchmarks are lacking for programming languages like Python, Scratch, and C#. Furthermore, there is a scarcity of manually validated benchmarks encompassing array of programming languages. (6) The DFS technique [31] (S19) demonstrated superior performance on both benchmarks in terms of precision, recall, and F1-score, achieving F1-scores of 97.9% and 99.4%, respectively. Following closely, ASTNN [47] F1-scores of 93.8% and 95.5% on the BigClone-Bench and OJClone benchmarks, respectively. Conversely, CDPU [60] (S14) and PBCS [48] (S1) exhibited the lowest performance levels in precision, recall, and F1-score on both benchmarks when compared to other techniques. It is worth noting that Study (S4), which utilized OLSTM, delivered the most impressive results among the detection techniques, attaining an impressive F-score of 98.04% when applied to open-source datasets sourced from diverse programming language applications. The utilization of RNN techniques for code clone detection is a promising field requiring further research. Consequently, it is crucial to invest more research efforts into implementing RNN-based techniques to address the challenge of code clone detection. Our analysis presents the current state of this research, furnishing vital insights for developers, researchers, and stakeholders. We anticipate that these findings will inspire future advancements in the field of code clone detection through the application of RNN techniques.

## Supporting information

**S1 File. PRISMA checklist for systematic review process.**
(DOC)

**S1 Appendix. List of relevant studies and quality assessment scores.**
(PDF)

## Acknowledgments

The authors express their gratitude for the support of University of Peshawar in the development of this work.

## Author Contributions

**Conceptualization:** Fahmi H. Quradaa, Sara Shahzad.

**Formal analysis:** Fahmi H. Quradaa, Sara Shahzad.

**Investigation:** Fahmi H. Quradaa, Rashad S. Almoqbily.

**Methodology:** Fahmi H. Quradaa, Sara Shahzad.

**Project administration:** Sara Shahzad.

**Validation:** Fahmi H. Quradaa, Sara Shahzad, Rashad S. Almoqbily.

**Writing – original draft:** Fahmi H. Quradaa.

**Writing – review & editing:** Fahmi H. Quradaa, Rashad S. Almoqbily.

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
