## [Decision Letter · Decision Letter 0]

30 Oct 2023

PONE-D-23-28599A systematic literature review on the applications of recurrent neural networks in code clone researchPLOS ONE

Dear Dr. Quradaa,

Thank you for submitting your manuscript to PLOS ONE. After careful consideration, we feel that it has merit but does not fully meet PLOS ONE’s publication criteria as it currently stands. Therefore, we invite you to submit a revised version of the manuscript that addresses the points raised during the review process.

We look forward to receiving your revised manuscript.

Kind regards,

Jacopo Soldani

Academic Editor

PLOS ONE

Journal Requirements:

Reviewers' comments:

Reviewer's Responses to Questions

**Comments to the Author**

1. Is the manuscript technically sound, and do the data support the conclusions?

Reviewer #1: Partly

Reviewer #2: Yes

2. Has the statistical analysis been performed appropriately and rigorously? 

Reviewer #1: No

Reviewer #2: N/A

3. Have the authors made all data underlying the findings in their manuscript fully available?

Reviewer #1: Yes

Reviewer #2: Yes

4. Is the manuscript presented in an intelligible fashion and written in standard English?

Reviewer #1: Yes

Reviewer #2: Yes

5. Review Comments to the Author

Reviewer #1: The paper proposes a systematic review on the application of Recurrent Neural Network in code-clone research. At first, 2099 scientific papers have been obtained from 7 different online databases by using a proper research string. Then, they have been filtered by removing (1) the duplicates and (2) the papers do not satisfy the eligibility requirements on title/abstract defined by the authors. After the filters, 49 papers have been selected. Four more papers have been added by considering the references of the selected papers. All the selected studies (49+4=53) have been assessed by the authors by reading the full texts and assigning a score according to quality assessment criteria defined by the authors. At the end, 20 papers have been included in the analysis.

The main issue of the proposed work is the shallowness. The papers considered are not deepened in the review, making difficult to understand their approach. In my opinion, the analysis conducted (which counts how many papers use a particular RNN architecture, or a particular code representation) is not enough to accept the manuscript.

For example, most of the selected papers represent the code using its AST. However, most of them use RNN architectures that can handle only sequential data (AST are trees). Hence, it would be interesting to understand how they transform the AST into a sequence, and especially if they use different strategies.

Another example is on the clone types (which are never defined). An interesting analysis could be to consider the scores obtained by the approaches under study in the different types of clones. This could be helpful to show if there are tasks that are satisfactory resolved by RNN architecture, and especially if there are tasks that deserve more attention.

Lastly (and most importantly), how model selection/model assessment has been conducted in the selected papers is never considered. It is known that the results obtained by ML models is highly influenced by how model selection/model assessment are conducted. To this end, it is crucial to understand if the dataset considered have public split to perform model selection/model assessment, and if there are standard strategies (e.g. cross-validation, hold-out, double-cross validation). This is crucial especially for the RQ7: without using the same model assessment strategy, we cannot compare different approaches.

Other issue:

- In 6 database searches, the search has been performed only on the abstracts. Why has not the search been performed on the full text (as it has been done in the Spring Link Digital Library)? For example, I found this paper “AST-path Based Compare-Aggregate Network for Code Clone Detection” (10.1109/IJCNN52387.2021.9534099) that it is not considered in the review.

- The inclusion criteria “The study must focus on code clone detection using RNN techniques.” is vague. Are there papers which contain the search term in the abstract that does not employ RNN architectures? I suggest the author to better specify the meaning of this criteria.

- All the exclusion criteria (except for the last one) are the same of the inclusion criteria: in fact, they are the negation of the inclusion criteria. I think this is useless.

- The clone-detection type are not defined.

- The PDG acronym is never defined. Also, the acronym of the Control Dependency Graph should be (CDG) and not CFG

Reviewer #2: The paper presents a systematic review of Recurrent Neural Networks (RNNs) applications for code clone detection. The authors' approach is solid and aligns with established literature recommendations for the task. Furthermore, by focusing on RNNs, their work has significant novelty compared to existing systematic reviews on clone detection.

I have a minor concern regarding the terminology used. The authors distinguish between different recurrent models by calling "shallow" RNN the original recurrent architecture by Rumelhart et al. (1985) [3] and similarly "shallow" LSTM the original work by Hochreiter et al. (1997) [2]. The term "shallow" isn't commonly used in this context by the Machine Learning literature and might confuse readers. For instance, shallow RNNs might refer to a completely different model, e.g., Dennis et al. (2019) [1].

My primary concern is with the sources included in the literature review. The deep learning community predominantly focuses on and publishes at conferences such as Neurips, ICLR, ICML, or AAAI [4]. Despite the inclusion of "databases renowned for publishing research on deep learning applications," none of them contains the proceedings of these fundamental venues. While these conferences focus on methodological issues, they also widely cover deep learning applications. Therefore, the paper might have missed significant contributions to RNNs and their application to clone detection. For completeness, their inclusion is fundamental when dealing with Deep Learning applications. One possible way to include these sources in the systematic review would have been to include DBLP as a further database.

In conclusion, while the paper is well-written, rigorous, and significant, I recommend considering possible missouts from deep learning venues in a further revision before possible acceptance.

References:

[1]: Dennis, Don, et al. "Shallow RNN: accurate time-series classification on resource constrained devices." Advances in Neural Information Processing Systems 32 (2019).

[2]: Hochreiter, Sepp, and Jürgen Schmidhuber. "Long short-term memory." Neural Computation 9.8 (1997): 1735-1780.

[3]: Rumelhart, David E; Hinton, Geoffrey E, and Williams, Ronald J (Sept. 1985). [Learning internal representations by error propagation](https://apps.dtic.mil/dtic/tr/fulltext/u2/a164453.pdf). Tech. rep. ICS 8504. San Diego, California: Institute for Cognitive Science, University of California.

[4]: https://scholar.google.nl/citations?view_op=top_venues&hl=en&vq=eng_artificialintelligence

6. PLOS authors have the option to publish the peer review history of their article (what does this mean?). If published, this will include your full peer review and any attached files.

Reviewer #1: No

Reviewer #2: No

---

## [Author Response · Author response to Decision Letter 0]

12 Nov 2023

I would like to express my sincere gratitude to the Editor and the reviewers for their valuable feedback and comments.

Editor's Journal comments and responses

• Comment 1: [Please ensure that your manuscript meets PLOS ONE's style requirements, including those for file naming.]

Response : We would like to inform you that we have adhered to PLOS ONE's style requirements as outlined in the PLOS ONE style templates.

• Comment 2: [We note that you have stated that you will provide repository information for your data at acceptance. Should your manuscript be accepted for publication, we will hold it until you provide the relevant accession numbers or DOIs necessary to access your data. If you wish to make changes to your Data Availability statement, please describe these changes in your cover letter and we will update your Data Availability statement to reflect the information you provide.]

Response : We propose a modification to the Data Availability statement. Currently, all data are contained within our systematic literature review paper. Additionally, I have incorporated a Supporting Information file containing all pertinent studies and their corresponding quality assessment scores. Consequently, I kindly request a revision of the Data Availability statement to read, "All relevant data are within the paper and its Supporting Information file".

• Comment 3: [Please include captions for your Supporting Information files at the end of your manuscript, and update any in-text citations to match accordingly.]

Response : captions for the Supporting Information files have been included at the end of the revised manuscript. Additionally, We have updated the in-text citations in accordance with the journal's guidelines.

Reviewers comments and responses

Comments from Reviewer 1:

• Comment 1: [ The main issue of the proposed work is the shallowness. The papers considered are not deepened in the review, making difficult to understand their approach. In my opinion, the analysis conducted (which counts how many papers use a particular RNN architecture, or a particular code representation) is not enough to accept the manuscript.]

Response : We appreciate the reviewer’s insightful suggestion and agree that it would be useful to conduct a meta-analysis instead of sample descriptive analysis. However, It is widely acknowledged that comparing and assessing clone detection techniques poses significant challenges, primarily due to the diversity of subject systems, benchmark datasets, and the lack of standardized similarity measures [1]. In the course of our Systematic Literature Review (SLR), we observed that most of the selected studies utilized different datasets and evaluation metrics (heterogeneous studies), making it infeasible to combine their results for a meta-analysis [2]. Moreover, with only 20 selected studies (most of them are heterogeneous studies), the application of a meta-analysis, as cautioned by Paul and Barari [2] and Borenstein et al. [3], would be inappropriate and they recommended to employ SLR to synthesize research in that domain. Therefore, we decided to use a descriptive synthesis analysis, as recommended by Paul and Barari [2] and Borenstein et al. [3]. It is important to note that this chosen technique for synthesizing the extracted data has been employed in several prior SLRs within this research domain, as indicated by references [1], [4],[5],[6],[7], and [8]. Nevertheless, we acknowledge the important of addressing this limitation in the paper. In response, we have added a sentence in the revised manuscript on page 15, lines 297–301, exploring why we opted not to use meta-analysis. 

• Comment 2: [ For example, most of the selected papers represent the code using its AST. However, most of them use RNN architectures that can handle only sequential data (AST are trees). Hence, it would be interesting to understand how they transform the AST into a sequence, and especially if they use different strategies. ]

Response : We appreciate the insightful recommendation from the reviewer. It would indeed be interesting to explore how code representation techniques are transformed into formats suitable for Recurrent Neural Network (RNN) based techniques. However, for our current study, providing a detailed elaboration of the transformation process for code representations into RNN-compatible formats might slightly extend beyond the scope of our work. Our aim in this research question (RQ3) is to extract data related to the techniques employed for representing source code in the selected studies. This approach aligns with the methodology applied in other SLRs in this domain [5], [6], [7], and [8], where code representation techniques are presented without extensive details on the conversion mechanisms, as these details are comprehensively covered in the referenced studies.

Nonetheless, we acknowledge that this comment has highlighted an opportunity for future and more extensive research in this direction. We have incorporated a recommendation to this opportunity in the revised manuscript. Please refer to page 35 of the revised manuscript, lines 676–680.

• Comment 3: [ Another example is on the clone types (which are never defined). An interesting analysis could be to consider the scores obtained by the approaches under study in the different types of clones. This could be helpful to show if there are tasks that are satisfactory resolved by RNN architecture, and especially if there are tasks that deserve more attention.]

Response : Thank you for pointing this out. The reviewer is correct and we have taken this feedback seriously and made substantial revisions to address the issue. In the revised manuscript, we have added a background section that contains a sub-section to clarifying the definitions of clone types and cloning process. For detailed information, we kindly direct your attention to the revised manuscript, specifically on pages 5 and 6, lines 119–141.

In response to the concern about the analysis of the scores obtained by the approaches under study for different types of clones in RQ 2, we agree and have made updates. We have now included the performance scores for each clone type in each selected study in Table 11 and have also revised the relevant paragraphs in RQ2. Additionally, we have added Figure 10 to visually present the performance scores for each clone type in the selected studies, providing a more comprehensive overview and we updated the corresponding paragraphs. You can find this updates on pages 19,20, and 21, lines 368-400.

• Comment 4: [ Lastly (and most importantly), how model selection/model assessment has been conducted in the selected papers is never considered. It is known that the results obtained by ML models is highly influenced by how model selection/model assessment are conducted. To this end, it is crucial to understand if the dataset considered have public split to perform model selection/model assessment, and if there are standard strategies (e.g. cross-validation, hold-out, double-cross validation). This is crucial especially for the RQ7: without using the same model assessment strategy, we cannot compare different approaches. ]

Response : We are grateful for the reviewer's insightful suggestion, and we acknowledge the importance of demonstrating how model selection and assessment have been conducted in the selected studies. In response to this valuable comment, we have made several updates to address this concern.

First, It is widely acknowledged that comparing and assessing clone detection techniques presents significant challenges, primarily due to the diversity of subject systems, benchmark datasets, and the lack of standardized similarity measures [1]. Consequently, we have refined the analysis of Research Question 7 (RQ7) to focus exclusively on the comparative analysis of studies that utilize RNN applications for code clone detection that used similar benchmarks, specifically BigCloneBenchmark and OJClone, as well as similar evaluations metrics. This adjustment aligns with the methodologies found in the relevant SLRs in this research domain [5] and [6]. This refinement enables us to offer a comprehensive performance comparison of each RNN-based technique, as summarized in Table 17 on page 28 and figures 15 and 16 on page 30 within the revised manuscript.

Furthermore, we have incorporated details regarding the validation techniques used in these studies for model comparison and selection during their respective comparative analyses. This information can be found in a dedicated table, labelled as Table 19, located on page 32 within the revised manuscript. All updates in RQ7 can be found in the revised manuscript on pages 27-32, lines 497 to 614. Additionally, we have added a paragraph in the discussion section addressing the validation techniques in the revised manuscript on page 38, lines 747 - 755.

• Comment 5: [ In 6 database searches, the search has been performed only on the abstracts. Why has not the search been performed on the full text (as it has been done in the Spring Link Digital Library)? For example, I found this paper “AST-path Based Compare-Aggregate Network for Code Clone Detection” (10.1109/IJCNN52387.2021.9534099) that it is not considered in the review. ]

Response : You have raised an important point here. However, In practice, papers abstracts are designed to highlight the most important aspects of a paper, which can be useful for quickly determining the relevance of a paper to the research topic at hand. Notably, in Springer Link Digital Library, advanced search settings offer researchers the ability to perform searches based on specific criteria, including keywords, exact phrases, the presence of at least one word, exclusion of specific words, and searching within titles. Detailed information on these search options is available in the search tips page located at [https://rd.springer.com/searchhelp]. 

Due to certain limitations, we made the decision to search using the full text of papers, as this particular online library, Springer Link Digital Library, does not provide the option to search solely within paper abstracts. This decision is in alignment with the methodologies identified in relevant SLRs within this research domain [1], [5], and [7]. A summary of the search process settings for each digital library is provided in the revised manuscript in Table 3 on page 11 to facilitate the reproducibility of this process.

In the paper titled 'AST-Path Based Compare-Aggregate Network for Code Clone Detection' (DOI: 10.1109/IJCNN52387.2021.9534099), the abstract presents a method for representing a code fragment as a collection of compositional paths within its abstract syntax tree (AST). This representation is used for training a machine learning (ML) classifier to identify code clone pairs. However, it is evident from the paper's abstract that its primary focus is on the application of ML techniques for code clone detection. In our research, as indicated in our search string on page 9, lines 198-204, our specific emphasis is on RNN techniques rather than general ML techniques. Consequently, when we apply our search string to all online databases, this paper will be excluded from our search results.

• Comment 6: [ The inclusion criteria “The study must focus on code clone detection using RNN techniques.” is vague. Are there papers which contain the search term in the abstract that does not employ RNN architectures? I suggest the author to better specify the meaning of this criteria.]

Response : Thank you for pointing this out. The inclusion criteria, "The study must focus on code clone detection using RNN techniques," means that a study can be included in our analysis only if its main emphasis is on detecting code clones (identical or near-identical pieces of code) using RNN techniques.

In response to this review, we have consequently revised this inclusion criteria to emphasize this point. Please refer to revised manuscript on page 11, lines 238-239. 

• Comment 7: [ All the exclusion criteria (except for the last one) are the same of the inclusion criteria: in fact, they are the negation of the inclusion criteria. I think this is useless.]

Response : We appreciate the reviewer for pointing this out. Upon a thorough review of the inclusion and exclusion criteria in our SLR, we have identified only a single criterion within the inclusion criteria, specifically the first criterion, is the opposite of the first criterion in the inclusion criteria. However, we want to highlight that this alignment is consistent with the established practices observed in prior SLRs [4], [5], [6], and [7]. In response to this feedback, we have carefully revised both the inclusion and exclusion criteria. For revised details, Please refer to revised manuscript on page 11, lines 238 to 248.

• Comment 8: [ The clone-detection type are not defined.]

Response : Thank you for pointing this out and we have taken action to address your comment. In the revised manuscript, we have added a background section that contains a sub-section to clarifying the definitions of clone types and cloning process. For detailed information, we kindly direct your attention to the revised manuscript, specifically on pages 5 and 6, lines 119–141.

• Comment 9: [ The PDG acronym is never defined. Also, the acronym of the Control Dependency Graph should be (CDG) and not CFG. ]

Response : Thank you for bringing this to our attention. We fully agree with your comment and we have taken action to address your concerns. As a result, we have revised the acronym of the Control Dependency Graph to (CDG), which we believe is now clearer. You can find this change on page 21 of the revised manuscript, in the lines 407-408.

Additionally, we have made further revisions to include the acronyms of the control flow graph (CFG) and program dependency graph (PDG). You can find these updates in the same revised manuscript on page 22 specifically in lines 412 and 413.

Comments from Reviewer 2:

• Comment 1: [ I have a minor concern regarding the terminology used. The authors distinguish between different recurrent models by calling "shallow" RNN the original recurrent architecture by Rumelhart et al. (1985) [3] and similarly "shallow" LSTM the original work by Hochreiter et al. (1997) [2]. The term "shallow" isn't commonly used in this context by the Machine Learning literature and might confuse readers. For instance, shallow RNNs might refer to a completely different model, e.g., Dennis et al. (2019) [1]. ]

Response : Thank you for pointing this out. The reviewer is correct and we have taken this feedback seriously, making substantial revisions to correct the terminology used. In the revised manuscript, we have change the terminology 'shallow RNN (SRNN)' into 'Original RNN (ORNN)', 'Shallow LSTM (SLSTM)' into 'Original LSTM (OLSTM)', and 'Shallow GRU (SGRU)' into 'Original GRU (OGRU)' in all sections, tables , and figures. 

• Comment 2: [My primary concern is with the sources included in the literature review. The deep learning community predominantly focuses on and publishes at conferences such as Neurips, ICLR, ICML, or AAAI [4]. Despite the inclusion of "databases renowned for publishing research on deep learning applications," none of them contains the proceedings of these fundamental venues. While these conferences focus on methodological issues, they also widely cover deep learning applications. Therefore, the paper might have missed significant contributions to RNNs and their application to clone detection. For completeness, their inclusion is fundamental when dealing with Deep Learning applications. One possible way to include these sources in the systematic review would have been to include DBLP as a further database.]

Response : Thank you for your suggestion. Exploring the DBLP database, which serves as an online reference for bibliographic information on significant computer science publications, would have indeed been an interesting point. However, as you may be aware, the process of conducting a SLR requires strict adherence to a well-defined protocol. In our current work, we are strictly following the protocol and guidelines established by Kitchenham et al. [9]. These guidelines provide a comprehensive framework for the search process and the criteria for selecting study resources.

In the context of an SLR, it is essential to create a comprehensive search string that encompasses all terms relevant to the research questions. This search string should be consistently applied across all selected online databases. While attempting to utilize the DBLP database, we encountered an issue. Unfortunately, we were unable to apply our search string, located on page 9 lines 198-204, within its search engine. This limitation arises from the fact that the advanced search options in the DBLP database are still under development, as indicated on their FAQ page [https://dblp.org/faq/1474589.html]. This hinders our ability to effectively utilize our search string within this database. Furthermore, we observed that both the phrase search operator and the Boolean NOT operator were disabled due to technical issues, as specified in their search help page [https://dblp.org/faq/1474589.html]. Consequently, our search string could not be effectively utilized within the DBLP database.

In an effort to address this limitation, we conducted searches for the mentioned conferences, namely Neurips, ICLR, ICML, and AAAI, in an attempt to locate any missing resources. However, the protocol that we are following in this SLR recommends the utilization of online databases as the primary source of study.

References 

1. Rattan D, Bhatia R, Singh M. Software clone detection: A systematic review. Information and Software Technology. 2013;55(7):1165-99.

2. Paul J, Barari M. Meta-analysis and traditional systematic literature reviews—What, why, when, where, and how? 2022;39(6):1099-115.

3. Borenstein M, Hedges LV, Higgins JP, Rothstein HR. Introduction to meta-analysis: John Wiley & Sons; 2021.

4. Al-Shaaby A, Aljamaan H, Alshayeb M. Bad smell detection using machine learning techniques: A systematic literature review. Arabian Journal for Science and Engineering. 2020;45(4):2341-69.

5. Lei M, Li H, Li J, Aundhkar N, Kim D-KJJoS, Software. Deep learning application on code clone detection: A review of current knowledge. 2022;184:111141.

6. Manpreet K, Dhavleesh R. A systematic literature review on the use of machine learning in code clone research. Computer Science Review. 2023;47:100528.

7. Zakeri-Nasrabadi M, Parsa S, Ramezani M, Roy C, Ekhtiarzadeh M. A systematic literature review on source code similarity measurement and clone detection: Techniques, applications, and challenges. Journal of Systems and Software. 2023;204:111796.

8. Ain QU, Butt WH, Anwar MW, Azam F, Maqbool B. A Systematic Review on Code Clone Detection. IEEE Access. 2019;7:86121-44.

9. Kitchenham B, Pearl Brereton O, Budgen D, Turner M, Bailey J, Linkman S. Systematic literature reviews in software engineering – A systematic literature review. Information and Software Technology. 2009;51(1):7-15.

---

## [Decision Letter · Decision Letter 1]

20 Dec 2023

A systematic literature review on the applications of recurrent neural networks in code clone research

PONE-D-23-28599R1

Dear Dr. Quradaa,

We’re pleased to inform you that your manuscript has been judged scientifically suitable for publication and will be formally accepted for publication once it meets all outstanding technical requirements.

Kind regards,

Jacopo Soldani

Academic Editor

PLOS ONE

Additional Editor Comments (optional):

Reviewers' comments:

Reviewer's Responses to Questions

**Comments to the Author**

1. If the authors have adequately addressed your comments raised in a previous round of review and you feel that this manuscript is now acceptable for publication, you may indicate that here to bypass the “Comments to the Author” section, enter your conflict of interest statement in the “Confidential to Editor” section, and submit your "Accept" recommendation.

Reviewer #1: All comments have been addressed

Reviewer #2: All comments have been addressed

2. Is the manuscript technically sound, and do the data support the conclusions?

Reviewer #1: Yes

Reviewer #2: Yes

3. Has the statistical analysis been performed appropriately and rigorously? 

Reviewer #1: Yes

Reviewer #2: N/A

4. Have the authors made all data underlying the findings in their manuscript fully available?

Reviewer #1: Yes

Reviewer #2: Yes

5. Is the manuscript presented in an intelligible fashion and written in standard English?

Reviewer #1: Yes

Reviewer #2: Yes

6. Review Comments to the Author

Reviewer #1: The authors have addressed all the comments raised in previous review. I do not have any further comments.

Reviewer #2: The authors addressed the concerns I had in my review. They sufficiently clarified the terminology for recurrent models in the manuscript. Additionally, their explanation regarding the absence of main Machine Learning venues due to adherence to a well-defined protocol and technical limitations within the DBLP database is well-founded. It might still be beneficial to briefly mention this reasoning within the main body of the manuscript for transparency and completeness.

7. PLOS authors have the option to publish the peer review history of their article (what does this mean?). If published, this will include your full peer review and any attached files.

Reviewer #1: No

Reviewer #2: No

---

## [Editor Report · Acceptance letter]

25 Jan 2024

PONE-D-23-28599R1 

PLOS ONE

Dear Dr. Quradaa, 

I'm pleased to inform you that your manuscript has been deemed suitable for publication in PLOS ONE. Congratulations! Your manuscript is now being handed over to our production team.

Kind regards, 

on behalf of

Dr. Jacopo Soldani 

Academic Editor

PLOS ONE